# Nonmagnetic framboid and associated iron nanoparticles with a space-weathered feature from asteroid Ryugu

Yuki Kimura [1] ✉, Takeharu Kato[2], Satoshi Anada [2], Ryuji Yoshida[2], Kazuo Yamamoto [2], Toshiaki Tanigaki [3], Tetsuya Akashi[3], Hiroto Kasai[3], Kosuke Kurosawa [4,5], Tomoki Nakamura[6], Takaaki Noguchi [7], Masahiko Sato [8,9], Toru Matsumoto [7], Tomoyo Morita[6], Mizuha Kikuiri[6], Kana Amano [6], Eiichi Kagawa[6], Toru Yada [9], Masahiro Nishimura[9], Aiko Nakato [10], Akiko Miyazaki [9], Kasumi Yogata [9], Masanao Abe[9], Tatsuaki Okada [9], Tomohiro Usui [9], Makoto Yoshikawa[9], Takanao Saiki[9], Satoshi Tanaka[9], Fuyuto Terui[11], Satoru Nakazawa [9], Hisayoshi Yurimoto [12], Ryuji Okazaki[13], Hikaru Yabuta [14], Hiroshi Naraoka[13], Kanako Sakamoto [9], Sei-ichiro Watanabe [15], Yuichi Tsuda[9] & Shogo Tachibana [8,9]

Extraterrestrial minerals on the surface of airless Solar System bodies undergo gradual alteration processes known as space weathering over long periods of time. The signatures of space weathering help us understand the phenomena occurring in the Solar System. However, meteorites rarely retain the signatures, making it impossible to study the space weathering processes precisely. Here, we examine samples retrieved from the asteroid Ryugu by the Hayabusa2 spacecraft and discover the presence of nonmagnetic framboids through electron holography measurements that can visualize magnetic flux. Magnetite particles, which normally provide a record of the nebular magnetic field, have lost their magnetic properties by reduction via a high-velocity (>5 km s$^{-1}$) impact of a micrometeoroid with a diameter ranging from 2 to 20 μm after destruction of the parent body of Ryugu. Around these particles, thousands of metallic-iron nanoparticles with a vortex magnetic domain structure, which could have recorded a magnetic field in the impact event, are found. Through measuring the remanent magnetization of the iron nanoparticles, future studies are expected to elucidate the nature of the nebular/ interplanetary magnetic fields after the termination of aqueous alteration in an asteroid.

The surfaces of small bodies of the Solar System without atmospheres are altered over time by exposure to solar wind and by micrometeoroid bombardments[1]: this process is called 'space weathering'. Examinations of the resultant traces of space weathering can potentially provide a detailed understanding of interplanetary processes such as relative ages of surfaces on airless bodies and accurate interpretation of remote sensing data[2,3]. However, most meteorites are composed of materials originating from the interiors of asteroids. These materials have therefore not been subjected to space weathering on the asteroids' surfaces. Exceptions are meteorites called

regolith breccias, which contain materials once present on the surfaces of asteroids[4]. Some of these meteorites have constituents that exhibit space-weathering textures[5]. However, such components have been subjected to heating during the lithification of the regolith breccias, implying that, to study space-weathering effects, we need to use retrieved samples that have been gently collected from the surfaces of extraterrestrial bodies by a spacecraft.

In the case of samples retrieved from the asteroid Itokawa, exposure of the silicate particles to the solar wind has been shown to have caused the formation of metallic iron particles on their surfaces, reducing their spectral reflectance[6]. In addition, analyses of samples brought back by the Hayabusa2 spacecraft from the C-type asteroid Ryugu have clearly shown that the alterations caused by space weathering differ from those of samples of S-type asteroids[2,3]. C-type asteroids experienced aqueous alteration and, therefore, contain large amounts of aqueous products such as phyllosilicates, magnetite, and carbonates. Ryugu also experienced large-scale reactions of minerals with water as a result of melting of ice because of an increase in its internal temperature after the formation of the parent body[7–10]. The parent body was subsequently destroyed by a collision with another small Solar System object, and the debris reaccumulated to form Ryugu[8].

Initial analyses of samples from Ryugu showed that the hydroxy groups of the phyllosilicate on the surface of the particles had been lost, and that solar-wind-induced space weathering had led to the reduction of $Fe^{3+}$ to $Fe^{2+}$ and to dehydration[2,11]. A remarkable dehydration reaction was observed in phyllosilicates partially melted by a micrometeoroid impact. Micrometeoroid impacts on the surfaces of small bodies occur randomly over time, and impact melts have been found in 1–2% of the particles in the Ryugu sample[2]. These analyses suggest that the weak absorption at 2.7 µm wavelength, which is attributed to an OH stretching vibration and is typically observed in the spectra of C-type asteroids, might represent the progression of space-weathering-induced dehydration on the surface of the asteroid rather than the amount of water in the asteroid's interior[2].

Therefore, the alteration effects of space weathering on the most abundant phyllosilicates on asteroid surfaces are gradually becoming clearer, whereas studies of the space weathering of magnetite, an important recorder of the nebular magnetic field, have been limited. Magnetite is universally found in carbonaceous meteorites as a major product of the aqueous alteration of asteroids during the early stages of the formation of the Solar System. Magnetite, a ferromagnetic mineral, is important as the major carrier of remanent magnetization. Measurements of the bulk remanent magnetization of meteorites can provide information on the intensity of the magnetic field at the time and place where the magnetite was formed during aqueous alteration in the corresponding parent body, enabling a discussion of the physical evolutionary process of planetary systems[12–20]. In the case of the Ryugu sample, measurements of the remanent magnetization of two fragments have indicated the presence of a nebular magnetic field of 41–390 µT when the magnetite was formed during aqueous alteration in the parent body of the asteroid[21]. On the other hand, no stable magnetization was observed in other three fragments[22]. Because most of the retrieved sample originated from the asteroid's surface, it is important to interpret the origin of remanent magnetization while taking into account its modification as a result of space weathering.

In this paper, we report that our electron holography studies of the magnetic fields of the framboids in the retrieved sample show a reduction of magnetite and the formation of iron particles, likely as a result of micrometeorite bombardment.

## Results and discussion
### Typical framboid
In the stored sample, the main A0064 particle (~3 mm in size, mass 6.7 mg) was surrounded by many tiny particles that had likely detached from the surface of this particle [Supplementary Information (SI), Fig. S1A]. Two of these tiny particles, named FO007 ($170 \times 90$ µm²) and FO008 ($180 \times 100$ µm²), were mounted onto an indium plate (SI, Fig. S1B) for easier handling and for scanning electron microscopy (SEM) observation (SI, Fig. S1C, D). The tiny particles divided into many pieces when pressed, and each piece was assigned a branch number (SI, Fig. S1E, SIF). The fact that they were easily broken is consistent with the report that Ryugu particles are brittle[8]. Compositional analysis by SEM–energy-dispersive X-ray spectrometry (EDS) was performed on particle A0064-FO007-I ($72 \times 54$ µm²), which had a relatively high iron content on its surface and contained many spherical subparticles resembling framboidal magnetites (Fig. 1). Ultrathin sections of particle A0064-FO007-I with a thickness of 150 nm (Fig. 1A) were prepared by focused-ion-beam (FIB) machining for subsequent analysis (Methods).

The ultrathin section from position i (FIBi) contained approximately a dozen framboidal magnetite particles with sizes in the range 500–900 nm (Fig. 2A). The magnetic domain structure of these particles was examined by electron holography, and the magnetic domains in each of the particles were found to have concentric circular magnetic structures, typical of submicrometer magnetite particles (Fig. 2B)[23–26].

### Nonmagnetic framboid
Ultrathin sections taken from the neighboring position ii (FIBii) contained approximately ten framboidal particles of size 400–800 nm (Figs. 2C, SI, S2A), similar to the particles observed in sections taken from position FIBi. Surprisingly, electron holography revealed an absence of magnetic-domain structures and showed a homogeneous contrast, suggesting that the particles were nonmagnetic and differed markedly from magnetite (Figs. 2D, SI, S2B, C).

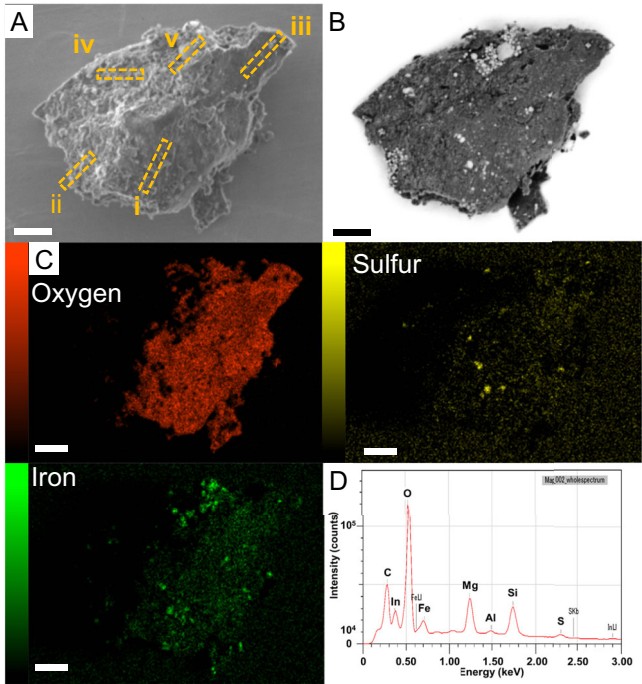

**Fig. 1 | Particle A0064–FO007–I of asteroid Ryugu. A** Secondary electron image. **B** Corresponding backscatter electron image. **C** Scanning electron microscopy–energy-dispersive X-ray spectrometry (EDS) elemental mapping at an acceleration voltage of 5 kV. Note that, because the detector was located to the right of the image, the left-hand side of the particle was shaded and could not be detected accurately. **D** EDS corresponds to the whole area of A0064-FO007-I in region (**C**). The scale bars for (**A–C**) are 10 µm.

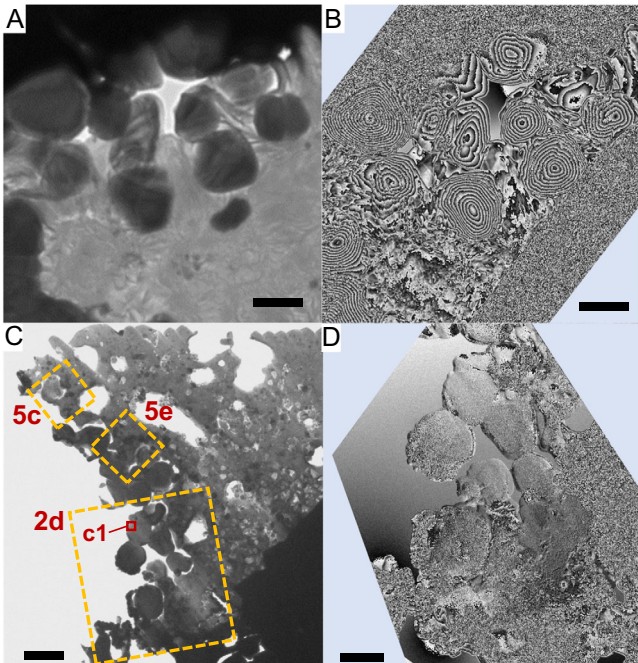

**Fig. 2 | Framboids in particle A0064–FO007–I of asteroid Ryugu. A, B**. Bright-field transmission electron microscopy (TEM) and the corresponding magnetic-flux-distribution images, respectively, of framboidal magnetite particles in a thin section prepared from Position i (FIBi) in Fig. 1A. **C** Bright-field TEM image of a thin section extracted from the Position ii (FIBii) in Fig. 1A. **D** Magnetic-flux-distribution image of box 2d in (**C**). All magnetic-flux-distribution images are two times the phase-amplified reconstruction. Scale bars are 500 nm for (**A**, **B**), 2 μm for (**C**) and 1 μm for (**D**).

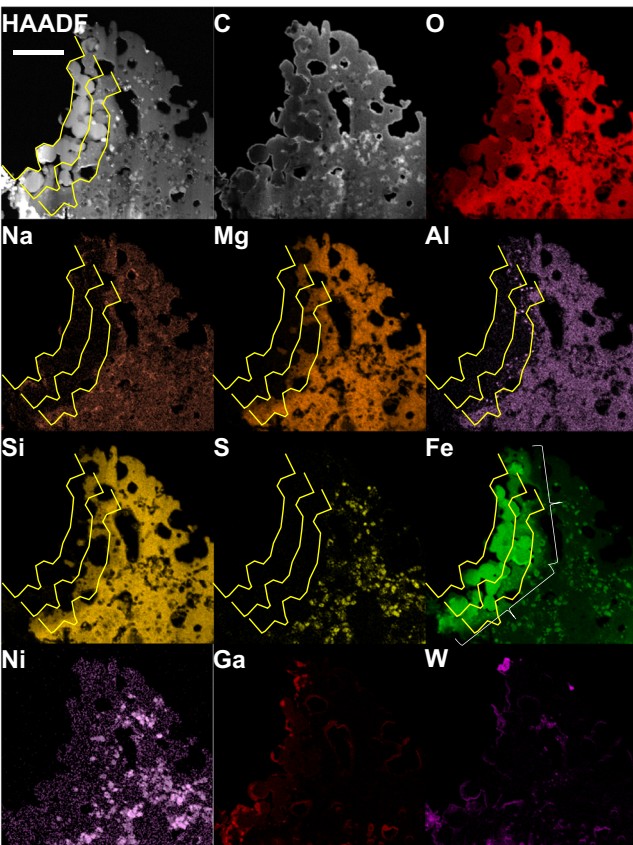

**Fig. 3 | High-angle annular dark-field–scanning transmission electron microscopy image and corresponding elemental mapping of the space-weathered framboid in FIBii.** This thin section corresponds to that in Fig. 1D. The original surface is at the left-hand side of the image. Note that 30-nm-thick carbon layers were deposited onto both faces of the thin section. Oxygen in the rounded particles at the surface and sodium, magnesium, silicon, and sulfur in the matrix just beneath the rounded particle have been depleted. Tiny aluminum round spots below the surface suggest melting at a high temperature. The yellow lines are guides, indicating the surface and positions 1 μm and 2 μm below the surface. The white brackets in the iron map indicate areas where iron appears to have diffused from the spherical particles into the surrounding matrix. The scale bar is 2 μm.

To investigate the composition of these nonmagnetic particles, scanning transmission electron microscopy (STEM)–EDS analysis was performed. The results show that the spherical particles contained iron and oxygen as the only major components, indicating that they consisted of iron oxides (Figs. 3, 4, SI, S3). However, the signal intensity for oxygen in the nonmagnetic particles compared with that in the matrix was weaker than that in typical magnetite, suggesting that the non-magnetic particles were formed by the reduction of magnetite. The average Fe/O ratio of the five spherical particles (SI, Fig. S3) was $0.728 \pm 0.035$, which is slightly smaller than that of magnetite ($Fe_3O_4$) at 0.75. However, the STEM–EDS results are not highly quantitative and are consistent with magnetite within the error range.

We therefore used electron energy-loss spectroscopy (EELS) to analyze the bonding state of iron and oxygen (Fig. 5A, B). In the EELS data of magnetite and maghemite, two characteristic peaks related to oxygen are generally observed at ~530 eV and ~540 eV, respectively, whereas the spectrum of wüstite shows a single peak at ~540 eV because of its relatively weak ~530 eV peak (Fig. 5A). A comparison of the spectra of magnetite, maghemite, and wüstite suggests that the nonmagnetic particles in Fig. 2D consist of wüstite because its spectrum shows only a single peak at 540 eV (c1 in Fig. 5A). However, the iron peak in the EELS data of wüstite is shifted toward lower energy compared with that in the spectra of magnetite and maghemite, whereas the position of the iron peak in the EELS data of our non-magnetic particles was closer to that of magnetite than to that of wüstite. Because wüstite is an antiferromagnetic mineral, the non-magnetic characteristics in the electron holography were consistent with the presence of wüstite. In summary, the nonmagnetic particles display characteristics of both magnetite and wüstite in terms of the bonding states between iron and oxygen. We have named such particles that display the characteristics of both wüstite and magnetite and that do not exhibit a magnetic-domain structure, 'pseudo-magnetite'.

On the left-hand side of each panel of the STEM–EDS mapping images of particle FIBii (Figs. 3, 4, SI, S3), corresponding to the surface of the A0064-FO007-I particle, not only is the signal for oxygen reduced in intensity, but the signal intensities for various other light elements are substantially reduced. Sodium and sulfur almost disappeared from the surface to the region ~2 μm below the surface. Magnesium and silicon also showed a compositional decline in the 1–2 μm range from the surface. Aluminum was present as numerous small spherical particles in the 1–2 μm region from the surface, suggesting that it might have melted and resolidified at one time. In addition, the elemental map of iron shows that the region in the matrix near the pseudo-magnetite particles was enriched in iron compared with the internal matrix. We believe that this iron had been released and diffused from the pseudo-magnetite.

## Metallic iron particles

Near the pseudo-magnetite particles in the same ultrathin section, vortex-like magnetic domain structures, characteristic of ferromagnetic materials, were observed (Fig. 5C–F). The EELS data of these particles showed an iron signal; however, the peak position was shifted to a lower energy compared with those of magnetite and wüstite (Fig. 5B). By contrast, the oxygen signal was very weak

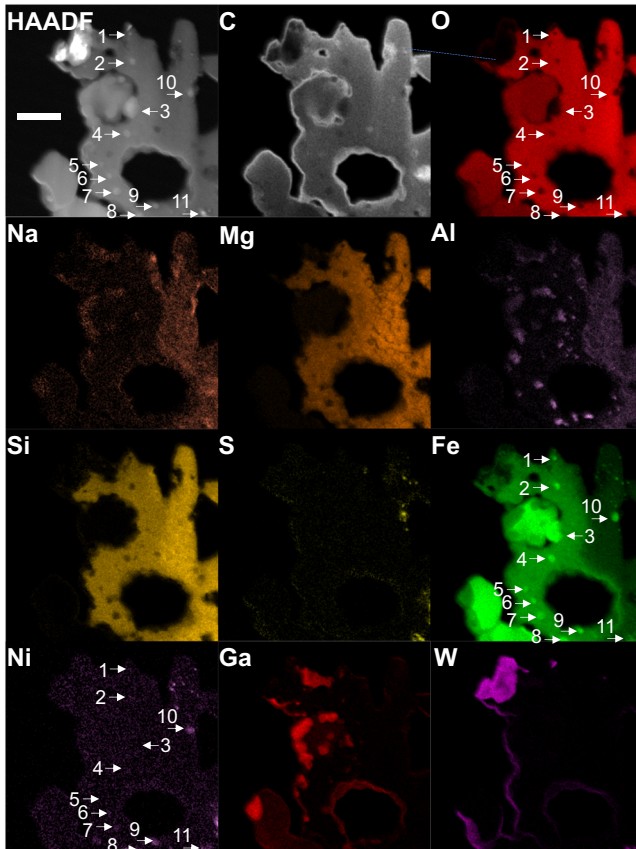

**Fig. 4 | Magnified high-angle annular dark-field–scanning transmission electron microscopy image of the pseudo-magnetite in FIBii and corresponding elemental mapping.** The eleven arrows indicate metals for which the elemental ratio of iron and nickel is shown in SI Table S1. The original surface is at the left-hand side of the image. Note that a 30-nm-thick layer of carbon was deposited onto each face of the thin section. The widespread distribution of iron in the matrix surrounding the pseudo-magnetite exclusively is likely a result of its release from the framboid. The scale bar is 500 nm.

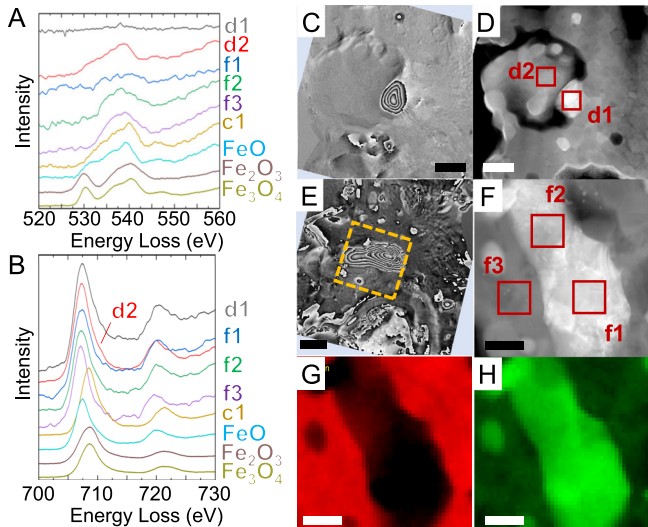

**Fig. 5 | Detail analysis of a framboid and surrounding iron particles. A, B.** Electron energy-loss spectroscopy (EELS) data of oxygen and iron, respectively, obtained from boxes c1 in Fig. 2, d2–f3. The bottom three spectra are reference spectra for synthesized wüstite, hematite, and magnetite (purity: 99.5%; Kojundo Chemical Lab. Co., Ltd.), respectively. **C, D** Magnetic-flux-distribution and high-angle annular dark-field (HAADF)–scanning transmission electron microscopy (STEM) images, respectively, for box 5c in Fig. 2. **E** Magnetic-flux-distribution image of box 5e in Fig. 2. **F** Magnified HAADF–STEM image of the box in (**E**). The STEM–energy-dispersive X-ray spectrometry elemental ratios in f1, f2, and f3 were Fe/Ni = 95.3:4.7, 98.0:2.0, and O/Mg/Al/Si/Fe = 52.7:13.6:0.5:17.0:16.3, respectively. **G, H** EELS maps of oxygen and iron, respectively, corresponding to (**F**). All magnetic-flux-distribution images are two times the phase-amplified reconstruction. Scale bars are 200 nm for (**C**–**E**) and 100 nm for (**F**–**H**).

or almost absent (Fig. 5A). The EELS maps of oxygen and iron (Fig. 5G, H) show that metallic iron particles were present. The combination of these results with those of STEM–EDS elemental mapping (Figs. 3, 4, SI, S3) suggest that the vortex particle in Fig. 5C consisted of metallic iron, whereas that observed in Fig. 5E consisted of an iron–nickel alloy containing 2−5 at% Ni. The magnetic vortex structures of the several-hundred-nanometer-sized iron particles are consistent with those in dusty olivines from the Semarkona meteorite[27]. The presence of many tiny iron particles without nickel near the surface of the region in Fig. 5C suggests that the tiny nickel-free iron particles formed as a result of the reduction of magnetite by micrometeoroid bombardment (Fig. 4, SI, Table S1). Since the iron particles in the interior contained 8−12 at% Ni, iron particles containing ~4 at% Ni in the intermediate region may be a result of reaction with the iron-nickel particles in the interior. Line profiles of iron and nickel from the surface to the interior are also consistent with this formation scenario involving impact heating (SI, Fig. S5).

More than one hundred metallic iron particles with a size of 30−400 nm were easily counted in the alteration region (shown by white brackets in Fig. 2 and arrows in SI, Fig. S5) 2 μm in depth from the surface, 10−20 μm in length, and 0.1 μm in thickness. Assuming that a single impact event alters a region 10 μm in diameter to a depth of 2 μm, the total number of iron particles produced would be on the order of ~$10^4$, which is a sufficient number to acquire remanent magnetization during an event involving the formation of iron

nanoparticles by micrometeoroid bombardment. The volume of the alteration region is estimated to be $(1.6−6.3) \times 10^{-16}$ m$^3$ by assuming a cylindrical shape, and the number density of the iron nanoparticles is the order of ~$10^{19}$ m$^{-3}$. The quantitative remanent intensity will be discussed briefly in Section "Future perspectives".

## Other types of pseudo-magnetite

The analysis results for ultrathin sections extracted from positions iii (FIBiii) and iv (FIBiv) of particle A0064-FO007-I are shown in Fig. 6. At position iii, a spherical particle ~2.5 μm in diameter and consisting of two distinctly different regions of contrast was isolated above the surface of the particle (Fig. 6A). Many microcrystals were found in the lower-contrast region at the top of the particle. The selected-area electron-diffraction (SAED) pattern of this region (circle b in Fig. 6A) shows that it was composed of polycrystalline wüstite and an amorphous material (Fig. 6B). STEM–EDS analysis indicates that this amorphous material was iron silicate (SI, Fig. S6). Intense spots in the SAED pattern corresponding to the particle's interior (Fig. 6C) indicate a single crystal of wüstite; weak diffraction spots attributed to magnetite were observed at the same time (Fig. 6C). Again, this particle was a pseudo-magnetite that exhibited features of both magnetite and wüstite. EELS was used to examine the bonding state between oxygen and iron (Fig. 6D). The oxygen peak at ~530 eV hardly appeared in the spectrum corresponding to the interior of the particle or in that corresponding to the pseudo-magnetite in Fig. 2. However, the peak position of iron was the same as that of wüstite and differed from that of the pseudo-magnetite in Fig. 2. Electron holography observation showed no magnetic domain structure characteristic of magnetite as well as the pseudo-magnetite in Fig. 2 (Fig. 6E). On the other hand, elemental mapping by STEM–EDS showed no evidence of light-element depletion or iron leaching (Figs. 6F, SI, S6). These results are not inconsistent with a scenario in which the particle was formed as a splash of a high-temperature melt heated by a micrometeoroid impact and then

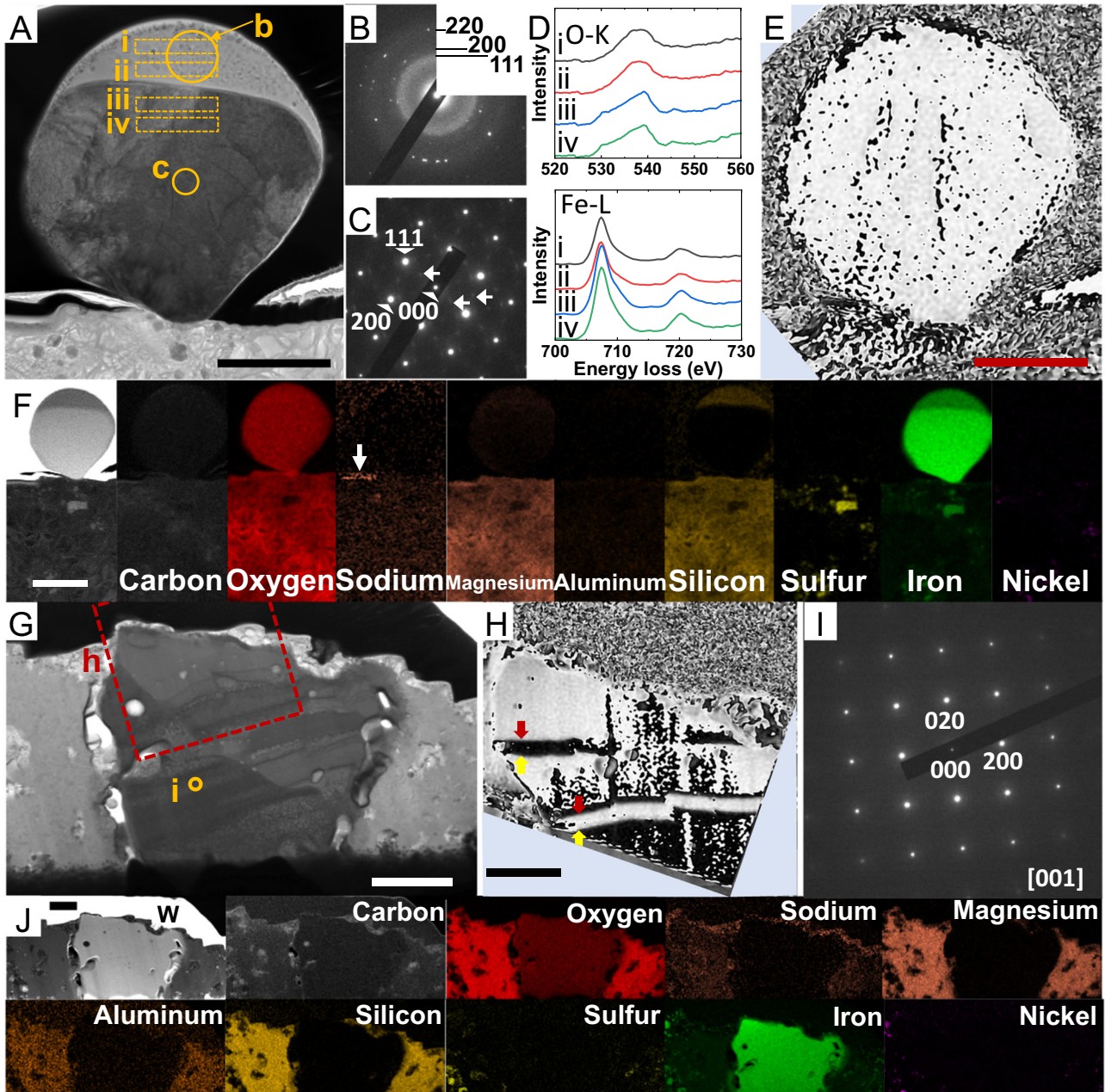

**Fig. 6 | Pseudo-magnetites. A, G** High-angle annular dark-field (HAADF)−scanning transmission electron microscopy (STEM) images of the thin sections extracted from Positions iii (FIBiii) and iv (FIBiv), respectively, of the surface of particle A0064−FO007−I, as shown in Fig. 1A. **B, C, I** Selected-area electron-diffraction patterns from b, c in (**A**), and i in (**G**), respectively. The indices in (**B, C, I**) correspond to FeO (JCPDS #6-0615). The broad ring in (**B**) originated from amorphous iron silicate. Weak spots in (**C**), indicated by white arrows, correspond to magnetite. **D** Electron energy-loss spectroscopy data of oxygen and iron obtained from the boxes i−iv in (**A**). **E, H** Magnetic-flux-distribution images corresponding to (**A**) and the square region in (**G**), respectively. **F, J** HAADF−STEM images and elemental mapping corresponding to (**A, G**), respectively. Although the white arrow in (**F**) indicates an intense signal for sodium, there was no characteristic peak in the energy-dispersive X-ray spectrometry data, suggesting an artificial contrast caused by the baseline. The entire data are given in SI, Figs. S6 and S8. The scale bars are 500 nm for (**A, E, I**) and 1 μm for (**F, G, J**).

solidified on the surface of the particle. The surface of Ryugu, which had been exposed to interplanetary space, is enriched in silicon compared with the interior of the asteroid[28]. The analysis of the ultrathin section FIBiii shows the same trend (SI, Fig. S7), indicating that the surface of this particle had also been exposed to interplanetary space.

In section FIBiv, we found a nonmagnetic iron oxide (Fig. 6G) that had no magnetic domain structure characteristic of magnetite (Fig. 6H). Its SAED pattern corresponded to that of wüstite taken from the <001> crystal zone axis (Fig. 6I). The surrounding matrix did not exhibit light-element depletion (Figs. 6J, SI, S8). The EELS data of oxygen and iron were consistent with that of wüstite; however, STEM−EDS analysis showed a higher Fe/O ratio ($1.45 \pm 0.05$) than that of wüstite, which might suggest further reduction from pseudo-magnetite (SI, Fig. S9). The external shape of this particle was not spherical; however, its interior contained voids surrounded by smooth surface features that appeared to have experienced a molten state. Because the light elements were not depleted and an iron-abundant region was not present around the pseudo-magnetite, we speculate that this wüstite might be the result of scattered magnetite entering the cavity of the matrix and solidifying (Figs. 6J, SI, S8).

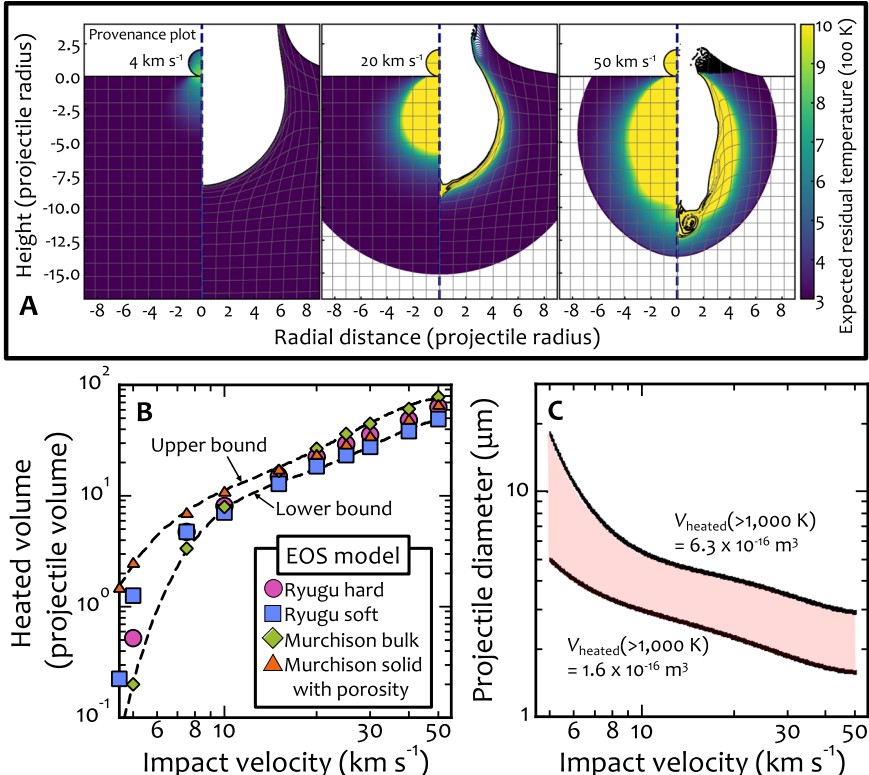

**Fig. 7 | Micrometeoroid size estimation. A** A provenance plot (left half) and a snapshot (right half) at the time when $t = 50\,t_s$, where $t$ is the time after the initial contact and $t_s$ is the characteristic time for projectile penetration defined by the projectile diameter divided by the impact velocity. The impact velocities are shown in each panel. The spatial scale is normalized by the projectile radius. The target is a flat surface. Residual temperatures in the region of maximum pressure <1 GPa are not calculated (white area). The yellow-colored regions indicate residual temperatures greater more than 1000 K. **B** Heated volume above 1000 K normalized by the projectile one as a function of impact velocity. We used the expected residual temperatures (see Methods) to estimate the heated volumes at given impact velocities. The results depend on the choice of the EOS models. Two dashed black lines are the upper and lower bounds. **C** The estimated projectile diameter as a function of impact velocity. We used the two dashed lines in the (**B**) in this calculation. The hatched region is an allowable range of the combination between the projectile diameter and impact velocity. The heated volume on the Ryugu particle is estimated to be $(1.6–6.3) \times 10^{-16}\,m^3$ by assuming a cylindrical shape with the diameter of $(10–20)\,\mu m$ and the thickness of $2\,\mu m$. The dimensions were determined with the microscopic images shown in Fig. 3.

## Possible space-weathering event

We surmised that pseudo-magnetite was created by reduction of magnetite as a result of space weathering by micrometeoroid impact. To confirm this hypothesis, we examined another sample, A0067, that had undergone space weathering over its entire millimeter-sized wide surface[2]. Because this sample had already been exposed to a strong magnetic field during conventional SEM observations in a previous study, the SEM mode associated with the FIB and conventional transmission electron microscopy (TEM) were used to search for framboids present on its surface (SI; Fig. S10A–E). Although sample A0067 displayed characteristic signatures of exposure to solar wind on the asteroid's surface[2], we did not observe a similar texture or pseudo-magnetite in its SAED patterns (SI, Fig. S10F–K), EELS data (SI, Fig. S11), or STEM–EDS elemental mapping (SI, Figs. S12, S13) of framboidal particles. In addition, heating an Orgueil meteorite sample to 773 K under the oxygen fugacity of the iron–wüstite buffer did not reduce the magnetite (SI, Fig. S14).

Magnetite coexists with wüstite and iron when reduced by hydrogen at 823–943 K[29]. In addition, the texture containing melted aluminum, as shown in Fig. 3, suggests that the sample had experienced heating to above the melting point of aluminum (933 K). Note that the aluminum signal is not inversely correlated with oxygen and, therefore, the texture may be alumina, which has a higher melting point than aluminum. On the other hand, pyrrhotite has been observed to remain stable at temperatures as high as 543 K in heating experiments[30]. The fact that iron sulfide and light elements such as sodium remained at depths greater than 3 μm from the surface of the sample (Fig. 3) suggests that an event occurred that heated the top 2 μm of the surface of

asteroid Ryugu to a temperature of ~1000 K. A previous laser-irradiation experiment using an Nd:YAG laser with a wavelength of 1064 nm and a pulse duration of 6–8 ns showed that Fe nanoparticles were produced from olivine grains on a very short timescale[31]. Short-duration laser irradiation can simulate high-speed collisions of micrometeoroids[32]. The spatiotemporal scale of this experiment is similar to that of the expected impact event in the present study. The pseudo-magnetites observed in the present study were not likely produced during the sampling operation by Hayabusa2 for the following reasons. The upper limit of the peak pressure experienced by Ryugu particles during sampling operations of Hayabusa2, in which a tantalum projectile was accelerated to $300 \pm 30\,m\,s^{-1}$ in the sampler horn[33], was estimated to be only 1.3 GPa[34]. These events were not sufficient to heat the Ryugu particles to 300 K[8]. Note that the Ryugu samples analyzed in this study were collected before the artificial cratering experiment on Ryugu[35]. These studies support the scenario in which the pseudo-magnetite observed in the present study was produced by a micrometeoroid bombardment.

To verify the space-weathering effect of micrometeoroid bombardment, we calculated the residual temperature based on shock-physics modeling using four different parameter sets for the materials of carbonaceous asteroids[8]. The snapshots (right half) shown in Fig. 7A suggest that the thickness of the high-temperature region is approximately equal to the diameter of the projectile, even at a high impact velocity of 50 km s⁻¹, and that the temperature rapidly decreases from the surface to the interior, reaching ~600 K at a thickness equal to approximately twice the diameter of the projectile at this time. The estimated range of projectile size and impact velocity was obtained by

comparing the volume heated above 1000 K as a function of the impact velocity (Fig. 7B) to the measured volume of the pseudo-magnetite layer (Fig. 7C). The projectile diameter must be smaller than the reduced region, i.e., 10–20 μm. With increasing impact velocity, the estimated projectile size decreases because the heated volume is constrained by the observations as the volume of the reduced region. Thus, even at an impact velocity of 50 km s$^{-1}$, the projectile size is expected to be ~2 μm. For Ryugu and a micrometeoroid to collide at a high velocity of >5 km s$^{-1}$, one of them must have an eccentric orbit.

The impact velocity distribution of micrometeoroids strongly depends on their origin (asteroids or comets) and the $\beta$ values, where $\beta$ is the ratio between the force due to solar-radiation pressure and that due to gravity. At least in the current Solar System, the velocity with which micrometeoroids with a radius of ~2 μm (~10$^{-13}$ kg), which corresponds to $\beta < 0.2$, impact asteroids located between 1.5 and 4.0 astronomical units (AUs) from the Sun can be greater than 10 km s$^{-1}$ [36] because of the radiation pressure. Ryugu formed in the outer part of the Solar System and may have already moved inward by the time of aqueous alteration[7]. Therefore, the micrometeoroid would have had a more eccentric orbit at the time of impact. The presence of a micrometeoroid with an eccentric orbit suggests not only that material from the inner Solar System diffused outward[37] but also that a large amount of material from the outer Solar System diffused inward. This is consistent with the finding of Fe$_4$N in the space weather layer of a Ryugu particle, which has been interpreted as a product of a reaction between metallic iron and ammonia that was provided by outer Solar System dust[3].

## Reduction sequences

There are nonmagnetic framboids that show different peak positions each other in the EELS data. A nonmagnetic framboid in box 5c of Fig. 2C has a peak at 707 eV (spectrum d2 in Fig. 5B), and a nonmagnetic framboid in box 2d of Fig. 2C has a peak at 708 eV (spectrum c1 in Fig. 5B) in the iron EELS data. The feature at 707 eV is in good agreement with those attributed to FeO (Fig. 5B) and Fe$_2$SiO$_4$[38], which are Fe$^{2+}$ reference materials. On the other hand, the feature at 708 eV is in good agreement with the feature attributed to Fe$_3$O$_4$ (Fig. 5B), which indicates that Fe$^{2+}$/Fe$^{3+}$ = 1/2. Because both are nonmagnetic framboids, the Fe$^{2+}$/Fe$^{3+}$ ratio does not appear to correlate directly with the presence of magnetism. However, oxygen spectra c1 and d2 are very similar (Fig. 5A). Both show the main peak at ~540 eV attributed to iron oxides without the pre-peak at ~530 eV observed in the spectra of Fe$_3$O$_4$ and Fe$_2$O$_3$. The lack of the pre-peak at 530 eV is consistent with the characteristics of FeO. Comparing the EELS peak positions of iron in the d2 region in Fig. 5D and the c1 region in Fig. 2C reveals that the framboid in the d2 region is more reduced than that in the c1 region because the c1 spectrum more closely resembles that of magnetite, whereas the d2 spectrum more closely resembles that of wüstite. That is, the nonmagnetic framboid in box 5c is closer to FeO than the nonmagnetic framboids in the box 2d region in Fig. 1C. In summary, the reduction of magnetite by a micrometeorite impact first results in the disappearance of the pre-peak in the EELS data for oxygen, followed by a decrease in the amount of Fe$^{3+}$ and a predominance of Fe$^{2+}$. The loss of magnetism is considered to occur with the disappearance of the oxygen pre-peak in the EELS data.

## Future perspectives

Magnetite and wüstite have similar cubic structures. The diffraction pattern of wüstite shows peaks at 0.249 (111), 0.215 (200), and 0.152 (220) nm ($hkl$ indices in brackets), and that of magnetite is similar, with peaks at 0.242 (222), 0.210 (400), and 0.148 (440) nm. Thus, magnetite and wüstite, especially as framboids, are difficult to distinguish from each other in terms of their crystal shape, composition, and conventional electron-diffraction pattern. However, a comparison of Fig. 2B, D shows that the magnetic properties of the two minerals are differ substantially and can assist in their detailed analysis. Pseudo-magnetite might account for only a few percent of the total magnetite.

Nevertheless, the detection of its presence will be important for precise interpretation when studying the paleomagnetism and microstructure of surface materials in meteorites and extraterrestrial samples retrieved by spacecraft such as OSIRIS-Rex, because pseudo-magnetite can serve as an indicator of specific alteration processes that the sample has undergone and its presence can be taken into account to adjust the interpretation of the magnetic history of the sample.

The iron particle precipitation associated with a micrometeoroid impact could play an important role as the remanence acquisition event for carbonaceous chondrites. Because iron particle precipitation is restricted in the thin surface layer of the impacted grain, the unaltered magnetite and pyrrhotite below the thin layer dominate the magnetic signal of grains larger than ~10 μm. However, the magnetic measurements for smaller grains would detect the remanence record of iron particle precipitated with the micrometeoroid impact. Moreover, the remanence component of metallic iron could be distinguished from those of magnetite and pyrrhotite by stepwise demagnetization treatments. The number of iron particles contained in the altered region is roughly estimated using the observed number density (10$^{19}$ m$^{-3}$) in altered area (10 μm in diameter and 2 μm in depth) as ~10$^4$. The 10$^4$ iron particles with a diameter of 100 nm had a total volume of ~5 × 10$^{-18}$ m$^3$ and a total saturation magnetization of ~9 × 10$^{-12}$ A m$^2$. The remanence intensity of the iron particles is estimated to be ~10$^{-13}$ A m$^2$, assuming that the intensity ratio of remanent magnetization to saturation magnetization is similar to that of thermoremanent magnetization of fine-grained magnetite acquired in a 100 μT field[39] and to be 0.01. Although the remanence intensity should depend on the external-field intensity and the remanence-acquisition process, the precipitation of iron particles could make the remanence detectable by a sensitive magnetometer, such as a scanning microscope with a superconducting quantum interference device[40]. In terms of statistical thermodynamics, the paleomagnetic samples should contain a sufficiently large numbers of ferromagnetic grains to obtain accurate paleomagnetic data[41]. An X-ray photoemission electron microscopy study on the Imilac and Esquel pallasite meteorites demonstrated that the regions containing ~10$^4$ grains recorded the systematic changes in a dynamo magnetic field[42], suggesting that the ~10$^4$ iron grains have sufficient ability to record paleomagnetic information. However, the estimation of the statistical error of paleomagnetism assuming specific conditions showed that the analyzed regions in these meteorites were not large enough to obtain accurate paleomagnetic data[41]. A number of collisions affecting a wider area or multiple events occurring during the timescale such that the external field could be regard as the same condition would give rise to a strong remanence intensity, resulting in increased detectability. The magnetic-field environment of the early Solar System has been reconstructed from the remanence records acquired as magnetite and pyrrhotite formed by aqueous alteration within asteroids[13–16,18,19,21], magnetite formed by igneous processes[20], and iron particles formed during chondrule formation[12,17,27]. The magnetic-field record of iron particles at the time of a collision event in a C-type asteroid could provide a spatiotemporally distinct constraint on the magnetic-field environment such as the dynamo magnetic field of an asteroid[43]. Note that, if the magnetite recording the natural remanence transformed into pseudo-magnetite/wüstite, the transformed magnetite particles does not contribute to the paleointensity estimation based on the remanence intensity ratio of natural remanence/laboratory remanence because the pseudo-magnetite/wüstite does not carry any remanences.

Micrometeoroid bombardments at a high impact velocity (>5 km s$^{-1}$) are expected to have commenced after the dissipation of the protoplanetary disk gas, because a high-velocity micrometeoroid impactor must have an eccentric orbit, which would cause its particles to be vaporized as a result of ablation by aerodynamic heating in the case of the presence of the disk gas. The iron nanoparticles surrounding the pseudo-magnetite therefore provide a record of the magnetic field after the dissipation of the disk gas. The dynamo

magnetic field of an asteroid is one important candidate of external field in such a period. In addition, sub-microscopic magnetite and associated metallic iron nanoparticles within iron sulfide were identified in lunar soil brought back by Chang'E-5[38]. The iron particles are thought to have been produced by eutectic reactions within oxygen-dissolved iron sulfide grains after reaction of molten iron sulfide with silicate gas during a large impact. The metallic iron particles were found to be magnetic. Thus, even after the dissipation of the disk gas, dynamo magnetic fields in small bodies may be imprinted during the impact of micrometeorites on the surface of airless bodies. A more comprehensive discussion of the formation epoch of nonmagnetic framboids and associated iron nanoparticles is expected in the future.

Micrometeoroid impacts similar to the one that we discovered in the present study are thought to have occurred frequently. However, these impacts have not been well understood in previous studies based on meteorites because the traces left by these events are only found on the uppermost surfaces of asteroids. This might be because, even if a meteorite fell to Earth with pseudo-magnetite or iron particles remaining, these particles would subsequently be oxidized by terrestrial weathering, although minor iron particles have been found in some carbonaceous chondrites despite them undergoing aqueous alteration[44]. In addition to our acquisition of the Hayabusa2 samples, retrieving samples from the asteroid Bennu by OSIRIS-REx would give us a chance to analyze them. At that time, we should avoid the remanence modification due to the oxidation of pseudo-magnetite and metallic iron nanoparticles.

## Methods
### Sample history
The Hayabusa2 spacecraft collected samples at two surface locations of the 162173 Ryugu asteroid[33]. The second sampling was conducted near the artificial crater produced by a small impactor carried by the spacecraft[35]. We principally analyzed Ryugu sample A0064, recovered during the first sampling operation (SI, Fig. S1A). Fragments of sample A0064 showed representative magnetic characteristics of the surface samples of Ryugu[21].

In a glove box, two fragments of A0064 were mounted by being placed on an indium plate using an ultrafine brush and then pressed with an SUS plate (SI, Fig. S1); this operation was performed at Tohoku University, Sendai, Japan. For our analyses, the samples were carried to the Japan Fine Ceramics Center, Nagoya, Japan, and stored in a glove box filled with argon. All sample preparations, sample transfer operations and analyses were performed in an atmosphere-free environment, except for a total of less than 15 s of exposure to the atmosphere when the samples were loaded into and unloaded from the tabletop scanning electron microscope. The samples on the indium plate were first coated with a 30-nm-thick layer of amorphous carbon using a carbon coater (PECS II; Gatan, Inc., Pleasanton, CA) to reduce the influences of electrical charging during SEM observations and FIB processing.

### Sample preparation
To identify framboidal magnetite, regions containing spherical particles with dominant iron and oxygen contents were selected on the basis of SEM images and elemental EDS maps, where a tabletop scanning electron microscope equipped with an EDS detector (JCM-7000, NeoScope; JEOL Ltd., Tokyo) was used under a 5 kV acceleration voltage to minimize the magnetic field (<50 μT). The sample was then placed in the FIB system (NB5000; Hitachi High-Tech Corp., Tokyo), and its surface was protected by tungsten deposited onto the processed area (Positions i–v in Fig. 1A). Thin sections were prepared by etching with Ga⁻ ion beam, and each section was mounted onto a molybdenum TEM grid. Unfortunately, the thin section taken from Position v (FIBv) was lost during the FIB processing. Other ultrathin sections were then prepared by further thinning of the mounted thin sections while the accelerating voltage was decreased from 40 to 5 kV under cryo-FIB conditions at −90 °C. In the final thinning, the ultrathin sections were tilted ±3

degrees to obtain a uniform thickness of 150 nm. Both faces of these ultrathin sections were coated again with carbon to a thickness of 30 nm to reduce the influence of electrical charging during electron holography observation. The internal potential images (SI, Fig. S15) confirm that the thickness inhomogeneity due to FIB processing does not cause artifacts in the magnetic-flux-distribution images.

### Sample analysis
Electron holography observations were performed with a specially designed holography transmission electron microscope equipped with a magnetic-field-free sample stage (magnetic field less than 17 μT) (HF-3300EH; Hitachi High-Tech Corp., Tokyo) operated at an acceleration voltage of 300 kV. Five holograms were taken in each observation area with an exposure time of 20 s. The reconstructed phase image of the electron wave passing through the sample contained information on an internal electric potential of the sample in addition to information on the magnetic flux. To subtract the internal potential, the sample was turned over and a second series of holograms were recorded from the other side of the thin section. By selecting one image from each of the five observed images from the front and back sides and subtracting the internal potential, we visualized the nanometer-scale magnetic domain structures and the magnetic-flux distribution of magnetic minerals with a spatial resolution of 14.4 nm. Similarly, we prepared internal potential images to evaluate the flatness of the ultra-thin section. For additional details of the image processing, refer to our previous report[23].

After the magnetic domain structures had been examined using the holography transmission electron microscope, STEM-EDS elemental mapping and electron-diffraction studies were carried out using an ordinary transmission electron microscope (JEM-F200, JEOL Ltd., Tokyo); EELS analysis was also performed using a JEM-ARM200 based Cs-corrected scanning transmission electron microscope (JEM-2400FCS, JEOL Ltd., Tokyo). All these analyses were performed at Japan Fine Ceramics Center.

### Computer simulations
We conducted shock-physics modeling to roughly constrain the size and impact velocity of micrometeoroids. We used the iSALE shock-physics code[45–47] iSALE-Dellen[48]. Two-dimensional cylindrical coordinates were employed. A projectile was divided into 50 cells per projectile radius (CPPR). This CPPR value is sufficiently high to enable accurate estimation of the peak-pressure distribution during an impact process[49]. The computational domain was set to 500 × 1500 cells in the radial ($R$) and vertical ($Z$) directions, respectively. We also set extension zones having 200 cells in the $R$ and $\pm Z$ directions with an extension factor of 2% to avoid interaction with reflected waves from the computational boundary. Therefore, the outcomes are free of effects from the choice of boundary conditions. The impact velocity ranged from 4 to 50 km s⁻¹. Because impact heating occurs at a very early stage of an impact event, we continued the numerical integration until the time in the computation when the characteristic time $t_s = D_p/v_{imp}$ reached 50, where $D_p$ is the projectile diameter and $v_{imp}$ is the impact velocity. In the calculations, we used the Tillotson equation of state[50] with four different parameter sets for the materials of carbonaceous asteroids[8]. The initial temperature was assumed to be 300 K, which is close to the current mean surface temperature of Ryugu[51]. Material strength and gravity were neglected in the simulations so that the results can be applied to any spatial scale. Lagrangian tracer particles were inserted into each computational cell.

We calculated the spatial distribution of post-shock residual temperature, which is the temperature after decompression as shown in Fig. 7A. We estimated the residual temperature as a function of the peak pressure, assuming an adiabatic expansion from the peak compression at the arrival of the compressive pulse. Thus, we stored peak pressures during the simulation on the tracers. Figure 7C shows the heated volume above 1000 K, which is normalized by the projectile volume, as

a function of the impact velocity. Because four EOS models provide a wide range of compressibility, the heated volume depends on the choice of the EOS model. The true values may lie in an intermediate range. Note that the "Murchison solid" model was combined with the $\varepsilon-\alpha$ porosity compaction model[47] to adjust the bulk density to the density of the Ryugu grain C0002[8]. We fitted the upper and lower envelopes of the data points with fourth-order polynomial functions to obtain the upper and lower limits pertaining to the heated volume. Figure 7C shows the estimated ranges of projectile size and impact velocity that cause the reduction of magnetites; the ranges were estimated by comparing the measured volume of the pseudo-magnetite layer with the detailed microscopic observation (see Fig. 3). The maximum size of the projectile must be the diameter of the reduced region (i.e., 10–20 μm). The estimated projectile size decreases as the impact velocity increases because the heated volume has been constrained by the observation as the volume of the pseudo-magnetite layer.

We here note currently unknown factors and their effects on the numerical results. The surface temperature might have been 100–200 K at the time of the collision, although we adopted the current surface temperature of Ryugu because Ryugu has been considered to have migrated from beyond the current Jupiter orbit to the current one during its history[8]. The expected residual temperature is roughly proportional to the initial temperature. The heated volumes above 1000 K shown in Fig. 7B are somewhat lower. Therefore, the estimated projectile diameter would be larger than that shown in Fig. 7C. The diameter is proportional only to one-third of the volume. In addition, a somewhat higher impact velocity can compensate for the change. Consequently, the effects of the initial temperature of the target body on the estimated projectile diameter are not significant.

The effects of material strength can be neglected with respect to hydrodynamic pressure during collisions at velocities greater than 10 km s$^{-1}$ [52]. At the lower impact velocities, the choice of the material parameter is important to accurately estimate the degree of impact heating in general because the material strength plays a major role in impact heating. Nevertheless, the effects of the material strength tend to reduce the contrast in the degree of heating between high-speed (>10 km s$^{-1}$) and low-speed (<10 km s$^{-1}$) collisions. In the case of the material strength, the estimated projectile diameter becomes somewhat smaller at <10 km s$^{-1}$ than the values shown in Fig. 7C: however, the value is still larger than that at >10 km s$^{-1}$. The main conclusion that the size of the micrometeoroid is 2–20 μm is not affected by the inclusion or exclusion of the material strength in the simulations.

We can neglect the effects of conductive cooling in the simulations as follows. On the basis of the thermal diffusivity of the Ryugu particle ($D_{diff} = 3.2 \times 10^{-7}$ m$^2$ s$^{-1}$)[8], the cooling timescale $t_{cool} = l_{mag}^2/D_{diff}$ pertaining to the pseudo-magnetite layer, which is $l_{mag} \sim 1$ μm in thickness, is estimated to be ~3 μs. By contrast, the time $t = 50\,t_s$ corresponds to 30 ns at $D_p = 3$ μm and $v_{imp} = 5$ km s$^{-1}$.

## Data availability
All data needed to evaluate the conclusions in the paper are present in the paper and/or the Supplementary Materials. The input files used in the iSALE simulation have been deposited in the Zenodo database and are accessible via https://zenodo.org/records/10423558. We used the EOS model in the iSALE simulations. The EOS data are taken from the JAXA Data Archives and Transmission System (DARTS) at https://data.darts.isas.jaxa.jp/pub/hayabusa2/paper/sample/Nakamura_2022.

## Code availability
The iSALE shock physics code was used to make Fig. 7. The iSALE code is not fully open-source; it is distributed on a case-by-case basis to academic users in the impact community for non-commercial use only. A description of the application requirements can be found at the iSALE website (https://isale-code.github.io). Any recent stable release can be used to reproduce the data shown in Fig. 7.

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

## Acknowledgements

The authors thank the Ministry of Education, Culture, Sports, Science and Technology (MEXT) (grants JPMXS0450200421, JPMXS0450200521) for financial support and JEOL for providing access to the tabletop scanning electron microscope (JCM-7000, NeoScope). This work was partly supported by JSPS KAKENHI Grant Number JP23H03981 (Y.K.). We thank the developers of iSALE, including G. Collins, K. Wünnemann, B. Ivanov, J. Melosh, and D. Elbeshausen. We also thank Tom Davison for the development of pySALEPlot. Numerical computations and analyses were in part carried out on the general-purpose PC cluster and the analysis servers at Center for Computational Astrophysics, National Astronomical Observatory of Japan.

## Author contributions

Y.K. and T.Na. designed the research; Y.K., T.K., S.A., R.Y., K.Ya., T.T., T.A., H.K., K.K., T.Na. and M.S. performed the research; Y.K., T.T., K.K., T.Na., T.No., M.S. and S.T. wrote the paper; and T.Na., T.No., T.Ma., T.Mo., M.K., K.A., E.K., T.Y., M.N., A.N., A.M., K.Yo., M.A., T.O., T.U., M.Y., T.S., S.Ta., F.T., S.N., H.Yu., R.O., H.Ya., H.N., K.S., S.W., Y.T. and S.Ta participated in the Hayabusa2 mission sample collection and curation.

## Competing interests

The authors declare no competing interests.

## Additional information

[1]Institute of Low Temperature Science, Hokkaido University, Sapporo 060-0819, Japan. [2]Nanostructures Research Laboratory, Japan Fine Ceramics Center, Nagoya 456-8587, Japan. [3]Research & Development Group, Hitachi, Ltd., Hatoyama, Saitama 350-0395, Japan. [4]Planetary Exploration Research Center, Chiba Institute of Technology, Narashino 275-0016, Japan. [5]Department of Human Environmental Science, Graduate school of Human Development and Environment, Kobe University, Kobe 657-8501, Japan. [6]Tohoku University, Sendai 980-8578, Japan. [7]Kyoto University, Kyoto 606-8502, Japan. [8]The University of Tokyo, Tokyo 113-0033, Japan. [9]ISAS/JAXA, Sagamihara 252-5210, Japan. [10]National Institute of Polar Research, Tashikawa 190-8518, Japan. [11]Kanagawa Institute of Technology, Atsugi 243-0292, Japan. [12]Hokkaido University, Sapporo 060-0810, Japan. [13]Kyushu University, Fukuoka 819-0395, Japan. [14]Hiroshima University, Higashi-Hiroshima 739-8526, Japan. [15]Nagoya University, Nagoya 464-8601, Japan. ✉e-mail: ykimura@lowtem.hokudai.ac.jp

