## [Peer Review File · Nature Communications]

Nonmagnetic frambooid and associated iron nanoparticles with a space-weathered feature from asteroid RyuguREVIEWER COMMENTS

Reviewer #1 (Remarks to the Author):

The authors unveil the discovery of nonmagnetic framboids and accompanying iron nanoparticles within the samples returned from the asteroid Ryugu. These unique findings are postulated to have originated from the reduction of magnetite during high-velocity impacts of micrometeorites after dissipation from the solar nebula. This research holds substantial interest and may offer valuable insights into comprehending the early evolution of material in the solar system. However, before acceptance for publication in Nature Communications, I recommend significant revisions.

1. The term "Space weathering" is conventionally employed to describe surface alterations on airless planetary bodies like the Moon, Mercury, and asteroids. It is essential to evaluate whether this term is appropriate for characterizing similar processes occurring after dissipation from the solar nebula.
2. In the Chang'e 5 lunar soil, sub-microscopic magnetite and associated metallic iron nanoparticles were identified (Guo et al., 2022). They are thought to have resulted from eutectic reactions within oxygen-dissolved iron-sulfide grains during large-impact events. While the sub-microscopic magnetite was confirmed to be nonmagnetic, the associated metallic iron particles were found to be magnetic. This discrepancy may be attributed, in part, to the recording of the solar wind magnetic field. Consequently, it is plausible that the solar wind magnetic field could also be imprinted during micrometeorite impacts on the surface of asteroid Ryugu, not solely during dissipation from the solar nebula. Therefore, a more comprehensive discussion regarding the formation era of nonmagnetic framboids and associated iron nanoparticles is warranted.
3. It is advisable to consider additional Electron Energy Loss Spectroscopy (EELS) measurements, with a particular focus on the Fe²⁺/Fe³⁺ ratio and the spatial distribution of Fe²⁺ and Fe³⁺ within the nonmagnetic framboids. These data are vital for confirming the presence of FeO and gaining insights into the reduction reaction of magnetite by micrometeorite impacts.
4. Line 177-179: The content of nickel in the metallic iron particles remains unclear. Typically, iron particles with a nickel content below 1% are assumed to result from iron leaching from magnetite during reduction induced by micrometeoroid bombardment.

Reference:

Zhuang Guo, Chen Li, Yang Li*, Yuanyun Wen, Yanxue Wu, Bojun Jia, Kairui Tai, Xiaojia Zeng, Xiongyao Li, Jianzhong Liu, and Ziyuan Ouyang, 2022, Sub-microscopic magnetite and metallic iron particles formed by eutectic reaction in Chang'E-5 lunar soil. Nature Communications, 13, 7177 (2022).
<https://doi.org/10.1038/s41467-022-35009-7>.

Reviewer #2 (Remarks to the Author):

Dear Kimura co-authors-

Below are my comments on the exciting and excellent manuscript 'Nonmagnetic framboid and associated iron nanoparticles with a space-weathered feature from asteroid Ryugu'. This manuscript presents an intriguing new insight into the process of space weathering through micro-meteorite impacts. Further, I am particularly excited by the application of electron holography to demonstrate the presence of non-magnetic phases. This is particularly powerful as the authors note that many iron minerals adopt a cubic structure and have very close chemical compositions. I am going to recommend 'Accepting with minor revisions'. My suggestions for revision are outlined in my comments below. Comments.

1) Figure 1 presents the central data of the manuscript. However, due to the complexity of the figure it can very hard to understand the impact of this figure with out multiple careful readings. Below are a few questions and comments to suggest improvements to the readability of what is the central figure of the manuscript.

a. Fig 1b is a Bright field image where as Fig 1D is a dark field image. The reversal in contrast is very confusing as it only becomes apparent that what is dark I the first image would be bright image. This is also only true if the reader is experienced with multiple S/TEM techniques, which is not necessarily the case for the very broad readership of Nat Comms. With that in mind does either a bright field image of Fig 1D exist, or a dark field of Fig 1b that they could be switched for?

b. Further along these lines, the effort to present the most complete data possible in a single figure means that the images in Fig 1h- 1m are very small. While many people will be reading this manuscript on a computer, I am wondering if this data is getting shortchanged by being squeezed into this figure. I would almost like to see these moved to the supplemental materials, which the authors are already using to great effect. This way Fig1b-g could become more prominent. I would also suggest moving Fig 1a to supplemental as well. By doing this Fig 1c could be made larger allowing for the interference fringes to better resolved. This would better demonstrate that while some magnetite grains exist and are magnetic, there exists a secondary population which is non-magnetic and made from wüstite.

c. Finally, for these frambooids after seeing the exciting more definitive SAED data in Fig. 3, I was wondering if this existed for the grains shown in Fig 1, and if it does exist could it be added into Figure 1?

2) Line 116 following places refers to an 'ultrathin section' prepared by FIB machining. This description of the subsamples is missing a critical parameter – the thickness of the 'ultrathin section'. While 100 nm can be acceptable for holography experiments, a simple estimate of the sample thickness allows the knowledgeable reader to better understand what the term 'ultrathin' means.

a. Related to this review of the Methods section describing sample preparation (lines 344-345) does not describe the methodology used to thin the sample. If a standard recipe was used this should be referred to in the references section. I would not expect a full detail, but key bits of information need to

communicated in relation to if low-kV final thinning was used and what tilts, since this impacts both the crystallinity and final thickness of the sample.

b. Related to this and what should be clearer from the description of thinning methods used above is a discussion of how parallel the final section is. While I do not expect a cross section of the section, it should be discussed that a sharply tapering sample (top edge closest to the tungsten cap is significantly thinner than the bottom edge) can lead to holography artifacts due to a non-uniform thickness. If the authors could expand on this in the methods section again it will help demonstrate that many of the holographic effects are not artifacts of thickness variations.

3) Lines 259- 260 provide a lower limit on the size of the micrometeorite(s?) (2 μm) which have impacted the surface of Ryugu. However, how the sentence is written leaves me wondering if there is there an upper limit on the size of these impactors? Can this be estimated from the model? I would imagine that beyond a critical size the impact effects of the micrometeorite would start inducing larger scale alterations to the Ryugu asteroid that the flash heating effects suggested by the numerical model. With this in mind it would make sense to state what the upper limit on the model would be.

4) Lines 283 -292 discuss the ability of a Scanning SQUID magnetometer to detect potentially weak remanence magnetisations in planetary samples. From the earlier work, do the authors have an estimate of how much of their sample is magnetite and how much would be wüstite? Building on this, can they use this estimate to suggest how much the transformation into pseudo-magnetite would reduce the measured paleo-intensity? These kinds of estimates would really help others in understanding discrepancies between the remanence signal that they are acquiring and the one that they would have expected.

5) Line 301 – The statement about when micrometeoroid bombardment starting shortly after the dissipation of the solar nebula begs a citation. I am not sure what this would be personally, but it the kind of knowledge that might be commonly accepted in a wide sector of the planetary sciences community, but for the more general readership it would be a statement which is not inherently apparent from the current text. Could you please add an appropriate reference?

Regards,

Reviewer #3 (Remarks to the Author):

Review of the manuscript by Kimura et al.

Summary of the key results

Kimura et al. present a broad microscopic and submicroscopic analysis of fragments of the asteroid

Ryugu material brought back by the Hayabusa 2 mission. This manuscript focuses on iron-rich minerals, putative magnetic carriers, contained in the fragments; in particular the putative effect of space weathering on these carriers is discussed. The authors identify various types of iron-bearing minerals, including magnetite framboids—confirming the results of previous studies—wüstite grains, iron metal grains, and a magnetite-like mineral they call “pseudo-magnetite”. Only the magnetite and iron grains are ferromagnetic according to their analysis. “Pseudo-magnetite” and wüstite are found at the surface of one FIB section on the fragments analyzed. They seem to correlate with an increased abundance of iron in the matrix, the presence of nanoscale iron metal grains, and aluminum droplets. The authors interpret these findings as evidence for a micrometeoroid impact that heated the first few microns of the samples. They propose a simple numerical modeling of the temperature gradient resulting from such an impact, showing that only a very high-velocity impact can explain the observed features on several μm inside the fragment. The authors conclude that the iron metal grains resulting from such an event could have recorded the intensity of the solar wind magnetic field as natural remanence.

Overall impression

The study is particularly detailed with a broad spectrum of experimental techniques. However, my impression is that the study is missing a quantitative analysis, for example of composition profiles for the minerals at stake, to support the formation scenario. The interpretation of the results in terms of magnetic remanence of the samples appears also far fetched in regards of already published studies.

Major questions/comments

I have four major comments regarding this manuscript.

“Pseudo-magnetite”

Although this is not my area of expertise, it is disturbing not to be able to identify this mineral. Could it be an intergrowth of magnetite and wüstite (explaining common EELS features with both minerals), as it is seen in impact spherules? That could explain why the magnetic-flux-distribution data do not show clear ferromagnetic signal. It would also be interesting to that have EELS spectrum of Fe for the comparison to be complete. Can the shift in Fe energy peaks be explained by reduction? I would refer to studies that show this. It would also be interesting to have a quantitative compositional profiles to confirm the qualitative observation that Fe leached out of the magnetite during impact heating. Is it seen in other meteorites? If so, it would be useful to refer to these studies. Also, how long would the impact heating last? Would that be enough for the iron to diffuse out of the magnetite?

Iron particles

The authors indicate that they can easily identify hundreds of submicron metal grains in the FIB section of Fig. 2. However, I could not identify where these grains are. I see the two iron grains identified in Fig. 1, but where are the hundreds of iron particles? On a more general note, figures should be complemented with marking to help visualize the features mentioned in the text. For the particles that is identified as an iron-nickel alloy, there is no quantitative compositional analysis. The composition is given out of nowhere, and it is a bit surprising to me that this would not be pure iron. It would be good to mention that minor iron particles can be found in carbonaceous chondrites (e.g., CM chondrites) despite them undergoing aqueous alteration.

Magnetic remanence

It is indicated in the text that the “quantitative remanent intensity” will be discussed in section 4 but there is no section 4. This whole part of the study is hard to follow. If I understand correctly, the authors conclude that the ferromagnetic iron particles could have acquired their remanence during their formation upon an impact of a micrometeorite. It is highly likely that the putative remanence would have been acquired after the dissipation of the nebula, and the author mention the solar wind magnetic field as putative source of magnetizing field. This idea is highly controversial. I would refer the author to a MHD study (Oran et al. 2019) that demonstrates that the pile up of the solar wind at the surface of chondritic body cannot explain remanence corresponding to a field higher than a few hundred of nT at best in the inner solar system (so weaker beyond the orbit of Jupiter). This is incompatible with published paleomagnetic results, but more importantly, it has been shown that magnetite and pyrrhotite largely dominate the magnetic properties of the returned samples. The significance of the remanence of a few iron grains inside a sample that contains abundant magnetite and pyrrhotite seems difficult to support. My opinion is that this whole part is too hand wavy and discredits other interesting results on the effect of space weathering on the mineralogy.

High-velocity impact

The velocity that is needed to reach aluminum temperature is, if I recall, very unlikely to be encountered—it would by the way be necessary to cite dynamical evolution studies to provide a distribution of the velocities of micrometeorites. I think an increased discussion about the feasibility of this scenario is needed. It is all the more needed that these pseudo magnetite formed during the impact, as well as the aluminum beads and volatile depletion, are not seen in other FIB sections if I am not mistaken. More details about the nature of the material employed in the simulations would be needed. In particular, material strength will largely change the temperature profile, no?

One related question: could the impact that excavated the particles during sampling be the cause of the observed features? This, I think, deserves to be discussed if rejected.

Precision and syntax

A lot of interpretations of observations in the manuscript are not based on precise arguments. I would suggest to back up each claims with either quantitative analysis (e.g., when you see a “compositional decline”, can you quantify it, is it significant?). I would also recommend to proof read again the manuscript, as there are a lot of typos and poor phrasing.

Point-by-point response to the reviewers' comments, reproduced verbatim

Reviewer #1 (Remarks to the Author):

The authors unveil the discovery of nonmagnetic frambooids and accompanying iron nanoparticles within the samples returned from the asteroid Ryugu. These unique findings are postulated to have originated from the reduction of magnetite during high-velocity impacts of micrometeorites after dissipation from the solar nebula. This research holds substantial interest and may offer valuable insights into comprehending the early evolution of material in the solar system. However, before acceptance for publication in Nature Communications, I recommend significant revisions.

Reply: We are grateful for your evaluation of the importance of our paper, and we appreciate your valuable comments. We have improved our paper by carefully following your suggestions. We have made our answers to your comments as clear as possible and have revised our manuscript accordingly. The changes in the article are indicated by yellow highlights in the manuscript and replies.

1. The term "Space weathering" is conventionally employed to describe surface alterations on airless planetary bodies like the Moon, Mercury, and asteroids. It is essential to evaluate whether this term is appropriate for characterizing similar processes occurring after dissipation from the solar nebula.

Reply: In general, material changes that occur on the surface of an airless body in relation to the external environment are referred to as space weathering. The presence of nebular gas would cause the micrometeorites to burn up as a result of friction with the gas, and cosmic rays and solar wind would not reach the asteroid, so space weathering does not occur. In this study, "space weathering" has been used for the process that occurred on the surface of the airless asteroid Ryugu.

2. In the Chang'e 5 lunar soil, sub-microscopic magnetite and associated metallic iron nanoparticles were identified (Guo et al., 2022). They are thought to have resulted from eutectic reactions within oxygen-dissolved iron-sulfide grains during large-impact events. While the sub-microscopic magnetite was confirmed to be nonmagnetic, the associated metallic iron particles were found to be magnetic. This discrepancy may be attributed, in part, to the recording of the solar wind magnetic field. Consequently, it is plausible that the solar wind magnetic field could also be imprinted during micrometeorite impacts on the surface of asteroid Ryugu, not solely

P

A

G

E

during dissipation from the solar nebula. Therefore, a more comprehensive discussion regarding the formation era of nonmagnetic framboids and associated iron nanoparticles is warranted.

Reply: Thank you for sharing the paper on the formation of iron particles by impact. In our study, non-magnetic spherical particles do not show a magnetic domain structure because they are no longer magnetite, whereas iron shows a magnetic domain structure because it is ferromagnetic. The presence of a domain structure does not depend on whether a magnetic field existed during the formation of these minerals. Of course, if a magnetic field was present, the iron particles could acquire a residual magnetic field; however, our analysis does not confirm this behavior. As you have pointed out, the nebular magnetic field might be recorded by tiny iron particles formed during the impact of micrometeorites on the surface of the asteroid Ryugu. Although it is not possible at this stage to determine the formation age of the nonmagnetic framboids and associated iron nanoparticles, the presence of microscopic iron particles of impact origin on the Moon is expected to extend to future research on the relationship between the formation of iron particles by impact and residual magnetic fields. We have added this point at the end of the third paragraph in “Future perspectives”.

3. It is advisable to consider additional Electron Energy Loss Spectroscopy (EELS) measurements, with a particular focus on the Fe²⁺/Fe³⁺ ratio and the spatial distribution of Fe²⁺ and Fe³⁺ within the nonmagnetic framboids. These data are vital for confirming the presence of FeO and gaining insights into the reduction reaction of magnetite by micrometeorite impacts.

Reply: It would certainly be useful to gain insight into the reduction reaction of magnetite on the basis of the Fe²⁺/Fe³⁺ ratio. Thank you. Reviewing our results again, we found two types of framboids: a nonmagnetic framboid with an Fe EELS peak at 707 eV [spectrum i2 in Fig. 1G (d2 in Fig. 5B in revision)] and a nonmagnetic framboid with a peak at 708 eV [spectrum e1 in Fig. 1G (c1 in Fig. 5B in revision)]. Although both are nonmagnetic framboids, the feature at 707 eV is in good agreement with the features of the Fe EELS data for FeO [Fig. 1G (Fig. 5 in revision)] and Fe₂SiO₄ (Guo et al., 2022), which are Fe²⁺ reference materials. On the other hand, the feature at 708 eV is in good agreement with the feature of the Fe EELS data for magnetite [Fig. 1G (Fig. 5 in revision)], which shows a Fe²⁺/Fe³⁺ ratio of 1/2. Thus, the Fe²⁺/Fe³⁺ ratio does not appear to correlate directly with the presence of magnetism. In the case of the EELS peaks, spectra i2 and e1 (d2 and c1 in revision) are similar, with both

P

A

G

E

showing the main peak at ~540 eV also observed in the spectrum of iron oxide, but without the pre-peak at ~530 eV observed in the spectra of Fe₃O₄ and Fe₂O₃. The lack of a 530 eV pre-peak is consistent with the characteristics of FeO. Comparing the EELS peaks of Fe in the i2 region in Fig. 1I (d2 region in Fig. 5D in revision) and the e1 region in Fig. 1D (c1 region in Fig. 2C in revision) reveals that, the i2 (d2 in revision) region is more reduced than the e1 (c1 in revision) region because the e1 (c1 in revision) spectrum is more similar to the spectrum of magnetite, whereas the i2 (d2 in revision) spectrum is more similar to the spectrum of FeO. That is, we can conclude that the nonmagnetic framboid in box h (5c in Fig. 2C in revision) is more similar to FeO than the nonmagnetic framboids in the box e (2d in Fig. 2C in revision) region in Fig. 1D (Fig. 2C).

In summary, the reduction of magnetite by micrometeorite impact first results in the disappearance of the pre-peak in the O EELS data, followed by a decrease of the signal intensity of Fe³⁺ and a predominance of Fe²⁺. The loss of magnetism is considered to occur with the disappearance of the O pre-peak in the EELS data. These discussions have been added in a new section "Reduction sequences".

4. Line 177-179: The content of nickel in the metallic iron particles remains unclear. Typically, iron particles with a nickel content below 1% are assumed to result from iron leaching from magnetite during reduction induced by micrometeoroid bombardment.

Reply: The nickel content in the iron particles near the surface, where the iron appears to leach out, was below the detection limit, whereas the iron particles in the interior contained ~10% nickel. In addition, iron particles that contained ~4% nickel were present in the middle region. These particles may have formed via reaction with the surrounding nickel. A description of these results has been added in section of "Metallic iron particles", along with a new Fig. 4, which is a modification version of the original Fig. S5, and new Table S1.

Reference:

Zhuang Guo, Chen Li, Yang Li*, Yuanyun Wen, Yanxue Wu, Bojun Jia, Kairui Tai, Xiaojia Zeng, Xiongyao Li, Jianzhong Liu, and Ziyuan Ouyang, 2022, Sub-microscopic magnetite and metallic iron particles formed by eutectic reaction in Chang'E-5 lunar soil. Nature Communications, 13, 7177 (2022). <https://doi.org/10.1038/s41467-022-35009-7>.

Reply: Thank you for clarifying the reference information, which is useful for the revision.

P
A
G
E

Reviewer #2 (Remarks to the Author):

Dear Kimura co-authors-

Below are my comments on the exciting and excellent manuscript 'Nonmagnetic framboid and associated iron nanoparticles with a space-weathered feature from asteroid Ryugu'. This manuscript presents an intriguing new insight into the process of space weathering through micro-meteorite impacts. Further, I am particularly excited by the application of electron holography to demonstrate the presence of non-magnetic phases. This is particularly powerful as the authors note that many iron minerals adopt a cubic structure and have very close chemical compositions. I am going to recommend 'Accepting with minor revisions'. My suggestions for revision are outlined in my comments below.

Comments.

Reply: We are grateful for your evaluation of the importance of our paper, and we appreciate your valuable comments. We have improved our paper by carefully following your suggestions. We have made our answers to your comments as clear as possible and have revised our manuscript accordingly. The changes in the article are indicated by yellow highlights in the manuscript and replies.

1) Figure 1 presents the central data of the manuscript. However, due to the complexity of the figure it can very hard to understand the impact of this figure with out multiple careful readings. Below are a few questions and comments to suggest improvements to the readability of what is the central figure of the manuscript.

Reply: The editor informed us that we could include as many as 10 figures in the text; we therefore followed your comments and made improvements.

a. Fig 1b is a Bright field image where as Fig 1D is a dark field image. The reversal in contrast is very confusing as it only becomes apparent that what is dark I the first image would be bright image. This is also only true if the reader is experienced with multiple S/TEM techniques, which is not necessarily the case for the very broad readership of Nat Comms. With that in mind does either a bright field image of Fig 1D exist, or a dark field of Fig 1b that they could be switched for?

Reply: HAADF-STEM image in Fig. 1D (Fig. 1C in revised version) has been changed to a

P

A

G

E

bright field TEM image to match Fig. 1B (Fig. 1A in the revised version).

b. Further along these lines, the effort to present the most complete data possible in a single figure means that the images in Fig 1h- 1m are very small. While many people will be reading this manuscript on a computer, I am wondering if this data is getting shortchanged by being squeezed into this figure. I would almost like to see these moved to the supplemental materials, which the authors are already using to great effect. This way Fig1b-g could become more prominent. I would also suggest moving Fig 1a to supplemental as well. By doing this Fig 1c could be made larger allowing for the interference fringes to better resolved. This would better demonstrate that while some magnetite grains exist and are magnetic, there exists a secondary population which is non-magnetic and made from wüstite.

Reply: Thank you for the suggestion. Original Figs. 1A and 1h-m have been moved to Figs. S2B and Fig. 2, respectively in the current version. Original Figs. 1E and 1F have also been moved to Fig. 2 in the current version. Original Figs. 1B-E have been moved to Figs. 1A-D in the current version to make the magnetic contour lines more clearly visible in contrast to the non-magnetic framboids.

c. Finally, for these framboids after seeing the exciting more definitive SAED data in Fig. 3, I was wondering if this existed for the grains shown in Fig 1, and if it does exist could it be added into Figure 1?

Reply: Actually, we cannot add SAED data to Fig. 1 because the FIB thin section dropped out just before the SAED data corresponding to Figs. 1C and D could be collected.

2) Line 116 following places refers to an ‘ultrathin section’ prepared by FIB machining. This description of the subsamples is missing a critical parameter – the thickness of the ‘ultrathin section’. While 100 nm can be acceptable for holography experiments, a simple estimate of the sample thickness allows the knowledgeable reader to better understand what the term ‘ultrathin’ means.

Reply: Ultra-thin sections were prepared to a thickness of 150 nm in this study. This information has been added in the last sentence of the first paragraph in section “Typical framboid”.

a. Related to this review of the Methods section describing sample preparation (lines 344-345)

P

A

G

E

does not describe the methodology used to thin the sample. If a standard recipe was used this should be referred to in the references section. I would not expect a full detail, but key bits of information need to be communicated in relation to if low-kV final thinning was used and what tilts, since this impacts both the crystallinity and final thickness of the sample.

Reply: The following sentences have been added in the third paragraph of the Methods section.

“Other ultrathin sections were then prepared by further thinning of the mounted thin sections while the accelerating voltage was decreased from 40 to 5 kV under cryo-FIB conditions at -90°C . In the final thinning, the ultrathin sections were tilted ± 3 degrees to obtain a uniform thickness of 150 nm.”

b. Related to this and what should be clearer from the description of thinning methods used above is a discussion of how parallel the final section is. While I do not expect a cross section of the section, it should be discussed that a sharply tapering sample (top edge closest to the tungsten cap is significantly thinner than the bottom edge) can lead to holography artifacts due to a non-uniform thickness. If the authors could expand on this in the methods section again it will help demonstrate that many of the holographic effects are not artifacts of thickness variations.

Reply: Because the internal potential of the sample is removed by taking the holograms from the front and back of the sample, the holographic image does not contain thickness information. On the other hand, the amount of integrated magnetic field changes depending on the thickness; thus, a quantitative discussion of holographic images should be conducted carefully. To show the thickness information of our sample, internal potential images have been added in Fig. S14. The contrast of the internal potential image is a function of the thickness of the sample under the assumption of uniform composition. This discussion has been added in the third paragraph in the Methods section.

3) Lines 259- 260 provide a lower limit on the size of the micrometeorite(s?) ($2\ \mu\text{m}$) which have impacted the surface of Ryugu. However, how the sentence is written leaves me wondering if there is there an upper limit on the size of these impactors? Can this be estimated from the model? I would imagine that beyond a critical size the impact effects of the micrometeorite would start inducing larger scale alterations to the Ryugu asteroid that the flash heating effects suggested by the numerical model. With this in mind it would make sense to state what the upper limit on the model would be.

P

A

G

E

Reply: The maximum size of the projectile must be the diameter of the reduced region, i.e., 10–20 μm . A more detailed discussion has been added in the third paragraph of section “Possible space-weathering event” and in the Methods section, in addition to new Figs. 7B and 7C.

4) Lines 283 -292 discuss the ability of a Scanning SQUID magnetometer to detect potentially weak remanence magnetisations in planetary samples. From the earlier work, do the authors have an estimate of how much of their sample is magnetite and how much would be wüstite? Building on this, can they use this estimate to suggest how much the transformation into pseudo-magnetite would reduce the measured paleo-intensity? These kinds of estimates would really help others in understanding discrepancies between the remanence signal that they are acquiring and the one that they would have expected.

Reply: Thank you for pointing out the unclear point concerning precise estimation of the contributions of magnetite reduction and iron particle formation. Because the magnetite reduction and iron particle precipitation are restricted in the thin surface layer of the original grain, the unaltered magnetite and pyrrhotite below the thin layer dominate the magnetic signal of meteoritic grain larger than $\sim 10 \mu\text{m}$. The effect of iron particles on paleointensity estimation is negligible in the case of large grains used in the previous analysis ($\sim 1 \text{ mm}$).

The paleointensity value is estimated on the basis of the ratio of natural remanence intensity to artificial remanence intensity. If the magnetite (ferromagnetic mineral) recording the natural remanence transformed into wüstite (non-magnetic phase), the intensities of both natural and artificial remanences for the transformed magnetite particles were subtracted from the above estimation; in such case, the subtraction does not contribute to the paleointensity estimation.

To clarify the above points, we have added descriptions of the relationship between grain size and magnetic signals in the second paragraph in section “Future perspectives”.

5) Line 301 – The statement about when micrometeoroid bombardment starting shortly after the dissipation of the solar nebula begs a citation. I am not sure what this would be personally, but it the kind of knowledge that might be commonly accepted in a wide sector of the planetary sciences community, but for the more general readership it would be a statement which is not inherently apparent from the current text. Could you please add an appropriate reference?

P

A

G

E

Reply: We appreciate the reviewer's constructive comment. Although we do not know appropriate references, the statement is based on a simple physical consideration. If a micrometeoroid has an eccentric orbit in the solar nebula, the particle must be ablated because of aerodynamic heating, similar to a meteor, resulting in vaporization. Therefore, the micrometeoroid bombardments began immediately after the dissipation of the solar nebula. We have inserted this explanation to the third paragraph in section "Future perspectives".

Reviewer #3 (Remarks to the Author):

Review of the manuscript by Kimura et al.

Summary of the key results

Kimura et al. present a broad microscopic and submicroscopic analysis of fragments of the asteroid Ryugu material brought back by the Hayabusa 2 mission. This manuscript focuses on iron-rich minerals, putative magnetic carriers, contained in the fragments; in particular the putative effect of space weathering on these carriers is discussed. The authors identify various types of iron-bearing minerals, including magnetite framboids—confirming the results of previous studies—wüstite grains, iron metal grains, and a magnetite-like mineral they call “pseudo-magnetite”. Only the magnetite and iron grains are ferromagnetic according to their analysis. “Pseudo-magnetite” and wüstite are found at the surface of one FIB section on the fragments analyzed. They seem to correlate with an increased abundance of iron in the matrix, the presence of nanoscale iron metal grains, and aluminum droplets. The authors interpret these findings as evidence for a micrometeoroid impact that heated the first few microns of the samples. They propose a simple numerical modeling of the temperature gradient resulting from such an impact, showing that only a very high-velocity impact can explain the observed features on several μm inside the fragment. The authors conclude that the iron metal grains resulting from such an event could have recorded the intensity of the solar wind magnetic field as natural remanence.

Overall impression

The study is particularly detailed with a broad spectrum of experimental techniques. However, my impression is that the study is missing a quantitative analysis, for example of composition profiles for the minerals at stake, to support the formation scenario. The interpretation of the results in terms of magnetic remanence of the samples appears also far fetched in regards of already published studies.

Reply: We appreciate your valuable comments. We have improved our paper by carefully following your suggestions. We have made our answers to your comments as clear as possible and have revised our manuscript accordingly. The changes in the article are indicated by yellow highlights in the manuscript and replies.

Major questions/comments

I have four major comments regarding this manuscript.

P

A

G

E

“Pseudo-magnetite”

Although this is not my area of expertise, it is disturbing not to be able to identify this mineral. Could it be an intergrowth of magnetite and wüstite (explaining common EELS features with both minerals), as it is seen in impact spherules? That could explain why the magnetic-flux-distribution data do not show clear ferromagnetic signal. It would also be interesting to that have EELS spectrum of Fe for the comparison to be complete. Can the shift in Fe energy peaks be explained by reduction? I would refer to studies that show this. It would also be interesting to have a quantitative compositional profiles to confirm the qualitative observation that Fe leached out of the magnetite during impact heating. Is it seen in other meteorites? If so, it would be useful to refer to these studies. Also, how long would the impact heating last? Would that be enough for the iron to diffuse out of the magnetite?

Reply: In the case of intergrowth of magnetite and wüstite, an oxygen pre-peak appears at 530 eV in the EELS data. The spectrum of pseudo-magnetite does not show any oxygen pre-peak; thus, the crystal structure of magnetite is not present. Also, if it is FeO, the EELS data for iron should show that all the iron is Fe²⁺; however, it also contains Fe³⁺. We understand your concern and have therefore shown all the data in the paper and look forward to future studies.

The shift in the iron energy peak can be explained by reduction. The best reference is the following paper mentioned by Reviewer 1:

Zhuang Guo, Chen Li, Yang Li*, Yuanyun Wen, Yanxue Wu, Bojun Jia, Kairui Tai, Xiaojia Zeng, Xiongyao Li, Jianzhong Liu, and Ziyuan Ouyang, 2022, Sub-microscopic magnetite and metallic iron particles formed by eutectic reaction in Chang'E-5 lunar soil. *Nature Communications*, 13, 7177 (2022).

<https://doi.org/10.1038/s41467-022-35009-7>.

Quantitative compositional profile data has been added in the new Fig. S5. No previous studies of iron leaching by the reduction of magnetite have been reported. However, sub-microscopic magnetite and associated metallic iron nanoparticles within iron sulfide were identified in lunar soil brought back by Chang'E-5. These results are described at the end of the third paragraph in section “Future perspectives”.

Estimating the time required for separation of iron nanoparticles from magnetites is difficult. Nevertheless, it is expected to be very short because the laser-irradiation experiment of Sasaki et al. (2001) clearly showed that iron nanoparticles

P

A

G

E

were produced from olivine grains under their conditions. They used a Nd:YAG laser with the wavelength of 1,064 nm and a pulse duration of 6–8 ns. The spatial and time scales of their experiments are similar to those considered in our study. Thus, the pseudo-magnetite observed in our study would also be produced by micrometeoroid impacts.

Reference: Sasaki, S., Nakamura, K., Hamabe, Y., Kurahashi, E., and Hiroi, T. (2001), Production of iron nanoparticles by laser irradiation in a simulation of lunar-like space weathering, *Nature* 410, 555–557.

Iron particles

The authors indicate that they can easily identify hundreds of submicron metal grains in the FIB section of Fig. 2. However, I could not identify where these grains are. I see the two iron grains identified in Fig. 1, but where are the hundreds of iron particles? On a more general note, figures should be complemented with marking to help visualize the features mentioned in the text. For the particles that is identified as an iron-nickel alloy, there is no quantitative compositional analysis. The composition is given out of nowhere, and it is a bit surprising to me that this would not be pure iron. It would be good to mention that minor iron particles can be found in carbonaceous chondrites (e.g., CM chondrites) despite them undergoing aqueous alteration.

Reply: The 100 metallic iron particles are indicated by arrows in Fig. S4. The composition ratios of iron to nickel for the 11 metallic particles shown in Fig. 4 are summarized in Table S1. As mentioned in the last paragraph of the section “Future perspectives”, minor iron particles can be found in carbonaceous chondrites despite them undergoing aqueous alteration.

Magnetic remanence

It is indicated in the text that the “quantitative remanent intensity” will be discussed in section 4 but there is no section 4. This whole part of the study is hard to follow. If I understand correctly, the authors conclude that the ferromagnetic iron particles could have acquired their remanence during their formation upon an impact of a micrometeorite. It is highly likely that the putative remanence would have been acquired after the dissipation of the nebula, and the author mention the solar wind magnetic field as putative source of magnetizing field. This idea is highly controversial. I would refer the author to a MHD study (Oran et al. 2019) that demonstrates that the pile up of the solar wind at the surface of chondritic body cannot explain remanence corresponding to a field higher than a few hundred of nT at best in the inner solar system (so weaker beyond the orbit of Jupiter). This is incompatible with published paleomagnetic results,

P

A

G

E

but more importantly, it has been shown that magnetite and pyrrhotite largely dominate the magnetic properties of the returned samples. The significance of the remanence of a few iron grains inside a sample that contains abundant magnetite and pyrrhotite seems difficult to support. My opinion is that this whole part is too hand wavy and discredits other interesting results on the effect of space weathering on the mineralogy.

Reply: Thank you for pointing out the unclear point in our manuscript. We discussed the remanence acquisition associated with iron precipitation as the possible magnetic field recording events for carbonaceous chondrites after the dissipation of the solar nebula. The candidates for magnetic field recording events include not only the solar wind but also the dynamo field in the parent asteroid. Oran et al. (2018) considered the average IMF component over the solar cycle, whereas the time scale of micrometeoroid impact is much shorter than that of the solar cycle. The inferred intensity of the amplified solar-wind field appears to be consistent with those of O'Brien et al. (2020) and Anand et al. (2022). To clarify the above points, we modified the descriptions (mainly second paragraph in section "Future perspectives").

We proposed the remanence of iron particles in the thin surface layer as the distinct magnetic-field record. To clarify this point, we added descriptions concerning the relationship between grain size and magnetic signals in the second paragraph of section "Future perspectives".

High-velocity impact

The velocity that is needed to reach aluminum temperature is, if I recall, very unlikely to be encountered—it would by the way be necessary to cite dynamical evolution studies to provide a distribution of the velocities of micrometeorites. I think an increased discussion about the feasibility of this scenario is needed. It is all the more needed that these pseudo magnetite formed during the impact, as well as the aluminum beads and volatile depletion, are not seen in other FIB sections if I am not mistaken. More details about the nature of the material employed in the simulations would be needed. In particular, material strength will largely change the temperature profile, no?

Reply: The impact velocity distribution of micrometeoroids strongly depends on their origin (asteroids or comets) and the β values, where β is the ratio of the force due to solar-radiation-pressure to that due to gravity. At least in the current Solar System, the impact velocities of the micrometeoroids with a radius of $\sim 2 \mu\text{m}$ ($\sim 10^{-13}$ kg), which

P

A

G

E

corresponds to $\beta < 0.2$, onto the asteroids located between 1.5 and 4.0 astronomical units (AUs) can be higher than 10 km s^{-1} [i.e., Altobelli et al., 2019] because of the radiation pressure. This discussion has been added in the last paragraph in section “Possible space-weathering event”.

We thank the reviewer’s careful review about the shock physics modelling. The effects of material strength can be neglected with respect to hydrodynamic pressure during the collisions higher than 10 km s^{-1} [Kurosawa and Genda, 2018]. At the lower impact velocities, the choice of the material parameter is important to accurately estimate the degree of impact heating in general since the material strength plays a major role in impact heating. Nevertheless, the effects of the material strength tend to reduce the contrast in the degree of heating between high ($>10 \text{ km s}^{-1}$) and low ($<10 \text{ km s}^{-1}$) speed collisions. In the case with the material strength, the estimated projectile diameter becomes somewhat smaller at $<10 \text{ km s}^{-1}$ than the values shown in Fig. 7C, but the value is still larger than that at $>10 \text{ km s}^{-1}$. The main conclusion that the size of the micrometeoroid is $2\text{--}20 \mu\text{m}$ is not affected with or without the material strength in the simulations. This explanation has been added in the ninth paragraph of the Methods section.

One related question: could the impact that excavated the particles during sampling be the cause of the observed features? This, I think, deserves to be discussed if rejected.

Reply: The samples we analyzed were recovered before the cratering experiment on the surface of the asteroid Ryugu; they were therefore not impacted.

The Hayabusa2 sampler accelerated a tantalum projectile to $300 \pm 30 \text{ m s}^{-1}$ in a sampler horn (Tachibana et al., 2022). Tomioka et al. (2023) estimated the peak pressure experienced by the Ryugu particles during the sampling operations to be 1.3 GPa. This shock compression is insufficient to heat the Ryugu particles to 300 K (Nakamura et al., 2022). Consequently, we can reject the possibility that the reduced iron oxides investigated in the present study are produced during the sampling operations. These discussions have been added in the second paragraph of section “Possible space-weathering event”.

Precision and syntax

A lot of interpretations of observations in the manuscript are not based on precise arguments. I would suggest to back up each claims with either quantitative analysis (e.g., when you see a “compositional decline”, can you quantify it, is it significant?). I would also recommend to proof

P

A

G

E

read again the manuscript, as there are a lot of typos and poor phrasing.

Reply: Quantitative discussion and analytical data (new Fig. S5 and Table S1) have been added and Fig. 4 has been modified to support our claim. If there are other critical places where quantitative analysis is missing, we would appreciate you pointing them out. The English of the paper has been improved by proof reading by a proof-reader:

P
A
G
E

REVIEWER COMMENTS

Reviewer #1 (Remarks to the Author):

The authors have responded well to my suggestions for revision, and the manuscript can be considered for publishing.

Yang Li
Institute of Geochemistry, CAS

Reviewer #3 (Remarks to the Author):

Second review of the manuscript by Kimura et al.

Thank you to the authors for taking into account my comments.

However, I still consider that there are some major comments to be made on the manuscript, in particular on the implications (“future perspective” – by the way isn’t this repetitive?) of the results.

Here are my main concerns for the section “future perspective”:

Paragraph #1

Why is the detection of “pseudo-magnetite” critical for the interpretation of paleomagnetic data given that it is non-ferromagnetic and therefore does not contribute to the remanence signal and cannot be remagnetized in the lab?

Paragraph #2

(1) How do you expect to measure the remanence of individual ~100 nm iron grains surrounded by magnetite grains? In dusty olivine chondrule studies, the chondrule is physically isolated and only the iron grains inside the chondrule contribute to the magnetic field measured with the SQUID microscope. Here, the signal would be polluted by the magnetic field of the neighboring magnetite. Isolating the rim containing iron metal blebs seems unrealistically hard to do, and for what gain (see my comment below)?

(2) I don’t understand what you mean by giving rise to a strong magnetization in line 364. First, doesn’t it seem unlikely that multiple micrometeorites would impact a nearby areas at the same time? Second, if you relax this constraints, then even if the grains recorded the solar wind, the latter would not have the same orientation, intensity between the two events and it is difficult to argue that magnetizing more grains would result in a stronger magnetization.

Paragraph #3

(1) The metal grains form after nebular dissipation, and, at least in Ryugu or Bennu samples, we do not expect a dynamo. You argue that the solar wind can reach instantaneous values of 1 μ T, but at what heliocentric distance? Many evidence suggest Ryugu formed far from the Sun: do you have any

constraints on the intensity of the solar wind at such large distance? You also do not cite the alternative paper suggesting that the solar wind is unlikely to give rise to a detectable magnetization in meteorites (Oran et al. 2018). Moreover, and this comes back to my previous comment, if more than one events are needed to give rise to a detectable magnetization of the iron grains, then this is unlikely that the putative solar wind magnetization would be cumulative. The claim that the CV and CM chondrites were magnetized by the solar wind is highly debated and you cannot mention this without mentioning the alternative claims.

(2) Since I am not convinced by your arguments in favor of a magnetizing solar wind and a detectable magnetization (using existing instruments) of the iron blebs magnetization, I find the conclusion that these measurements will help understand the composition of the solar wind and the early evolution of Venus' atmosphere seems out of place.

(3) The iron particles in Change'5 samples: are they magnetic or magnetized? What are the age of the soils? What field may they have recorded? What is the mineralogy of the samples? Is it really comparable to the magnetic mineralogy of Ryugu? I don't quite understand how this is a convincing argument to say that the magnetization of the iron grains in Ryugu samples is exploitable.

In addition, here are a few other comments:

- What are your arguments against a terrestrial alteration product during processing of the sample for the origin of the “pseudo-magnetites”? Isn't there no other iron oxides possible besides magnetite, maghemite and wüstite?
- L35: what signatures?
- L40: so they are not magnetite anymore, right?
- L41: to my knowledge, no paleomagnetic study of framboid-bearing extraterrestrial samples has unequivocally shown that framboids carry a non-zero magnetization. I would not use “normally”.
- L45: recorded a magnetic field, not acquired.
- L54: traces of what?
- How can an examination of the “traces” provide insight on interplanetary processes? Can you be more explicit?
- Aren't meteorites subjected to space weather after their excavation event?
- What is the lithification of the regolith breccias?
- L92: remove “as its name suggests”
- L92-94 I don't understand this sentence
- L97: this is IF the aqueous alteration occurs in the lifetime of the solar nebula field.
- L98-100: Note that in a recent publication, we propose a different interpretation of their results in light of measurements we conducted on other Ryugu samples. We do not find any stable magnetization in our 3 samples and argue that the results of Nakamura et al and Sato et al are due to magnetic contamination. This might affect some of your discussions in this manuscript. Ref: Maurel, C., J. Gattacceca, and M. Uehara. 2024. “Hayabusa 2 Returned Samples Reveal a Weak to Null Magnetic Field during Aqueous Alteration of Ryugu's Parent Body.” *Earth and Planetary Science Letters* 627 (February): 118559.
- L110: remove “surrounding”
- Table S1: what are the units? Are these normalized total? If so, can you indicated the non-normalized total ?
- L185: why is the Ni content not so well constrained?

- L190-194 and associated figure: hard to understand this part.
- L201: reference to back up that 10^4 particles are enough?
- L235-237: where are the associated data?

Point-by-point response to the reviewers' comments, reproduced verbatim

Reviewer #1 (Remarks to the Author):

Dear Prof. Yang Li,

Much appreciated your valuable comments and evaluation of the importance of our paper.

Sincerely,

Yuki

P
A
G
E

Reviewer #3 (Remarks to the Author):

Second review of the manuscript by Kimura et al.

Thank you to the authors for taking into account my comments.

However, I still consider that there are some major comments to be made on the manuscript, in particular on the implications (“future perspective” – by the way isn’t this repetitive?) of the results.

Here are my main concerns for the section “future perspective”:

Paragraph #1

Why is the detection of “pseudo-magnetite” critical for the interpretation of paleomagnetic data given that it is non-ferromagnetic and therefore does not contribute to the remanence signal and cannot be remagnetized in the lab?

Reply: The detection of pseudo-magnetite enriches the context in which paleomagnetic data is interpreted, providing a more subtle understanding of the history of the sample and the environmental conditions that formed its magnetic properties. This highlights the complexity of interpreting paleomagnetic signals and the importance of considering non-ferromagnetic as well as ferromagnetic components to accurately reconstruct the magnetic history of extraterrestrial samples. The sentence was rewritten as follows:

“Nevertheless, the detection of its presence will be important for precise interpretation when studying the paleomagnetism and microstructure of surface materials in meteorites and extraterrestrial samples retrieved by spacecraft such as OSIRIS-Rex, because pseudo-magnetite can serve as an indicator of specific alteration processes that the sample has undergone and its presence can be taken into account to adjust the interpretation of the magnetic history of the sample.

Paragraph #2

(1) How do you expect to measure the remanence of individual ~100 nm iron grains surrounded by magnetite grains? In dusty olivine chondrule studies, the chondrule is physically isolated and only the iron grains inside the chondrule contribute to the magnetic field measured with the SQUID microscope. Here, the signal would be polluted by the magnetic field of the neighboring magnetite. Isolating the rim containing iron metal blebs seems unrealistically hard to do, and for what gain (see my comment below)?

P

A

G

E

Reply: Thank you for letting us know the unclear point of manuscript. We do not claim to measure the remanent magnetization of individual iron particles. The remanence components of different magnetic minerals, in terms of composition, mineral phase, grain size and shape, can be generally distinguished by stepwise demagnetization treatments. To clarify this, we added the sentence: “Moreover, the remanence component of metallic iron could be distinguished from those of magnetite and pyrrhotite by stepwise demagnetization treatments.”

(2) I don't understand what you mean by giving rise to a strong magnetization in line 364. First, doesn't it seem unlikely that multiple micrometeorites would impact a nearby areas at the same time? Second, if you relax this constraints, then even if the grains recorded the solar wind, the latter would not have the same orientation, intensity between the two events and it is difficult to argue that magnetizing more grains would result in a stronger magnetization.

Reply: It depends on the timescale in which multiple events occur as you pointed out. We agree your opinion that the detection of remanence record of solar wind is difficult and deleted the descriptions concerning the remanence record of solar wind.

Revised sentence: “A number of collisions affecting a wider area or multiple events occurring during the timescale such that the external field could be regard as the same condition would give rise to a strong remanence intensity, resulting in increased detectability.”

Paragraph #3

(1) The metal grains form after nebular dissipation, and, at least in Ryugu or Bennu samples, we do not expect a dynamo.

Reply: We do not expect a dynamo magnetic field in Ryugu and Bennu, while we propose the remanence of metal grains as the recorder of a dynamo magnetic field in an asteroid. To clarify this, we added the descriptions concerning the dynamo magnetic field in lines 394-396 and 400-402.

You argue that the solar wind can reach instantaneous values of 1 μ T, but at what heliocentric distance? Many evidence suggest Ryugu formed far from the Sun: do you have any constraints on the intensity of the solar wind at such large distance?

Reply: We agree your opinion that the detection of remanence record of solar wind is difficult. We deleted the descriptions concerning the remanence record of solar wind.

You also do not cite the alternative paper suggesting that the solar wind is unlikely to give rise to

P

A

G

E

a detectable magnetization in meteorites (Oran et al. 2018).

Reply: We deleted the descriptions concerning the remanence record of solar wind.

Moreover, and this comes back to my previous comment, if more than one events are needed to give rise to a detectable magnetization of the iron grains, then this is unlikely that the putative solar wind magnetization would be cumulative.

Reply: We deleted the descriptions concerning the remanence record of solar wind.

The claim that the CV and CM chondrites were magnetized by the solar wind is highly debated and you cannot mention this without mentioning the alternative claims.

Reply: We deleted the descriptions concerning the remanence record of solar wind.

(2) Since I am not convinced by your arguments in favor of a magnetizing solar wind and a detectable magnetization (using existing instruments) of the iron blebs magnetization, I find the conclusion that these measurements will help understand the composition of the solar wind and the early evolution of Venus' atmosphere seems out of place.

Reply: We deleted the descriptions concerning the remanence record of solar wind.

(3) The iron particles in Change'5 samples: are they magnetic or magnetized? What are the age of the soils? What field may they have recorded? What is the mineralogy of the samples? Is it really comparable to the magnetic mineralogy of Ryugu? I don't quite understand how this is a convincing argument to say that the magnetization of the iron grains in Ryugu samples is exploitable.

Reply: Thanks, on your interest about the iron particles in Change'5 samples. All your questions are out of our scope. Here, we simply follow the comments of another reviewer and provide an example of metallic iron particles forming in extraterrestrial objects. We are not claiming that the magnetization of iron grains in the Ryugu sample is exploitable based on observations of iron particles on the Moon.

In addition, here are a few other comments:

– What are your arguments against a terrestrial alteration product during processing of the sample for the origin of the “pseudo-magnetites”? Isn't there no other iron oxides possible besides magnetite, maghemite and wüstite?

Reply: We all prepare our samples using the same analytical procedures. Nevertheless,

P

A

G

E

pseudo-magnetites were observed only in some of the samples. We also found no evidence of alteration during processing around the pseudo-magnetite.

We performed a detailed analysis using EELS and electron diffraction. No other iron oxides are possible.

– L35: what signatures?

Reply: “of space weathering” has been added.

– L40: so they are not magnetite anymore, right?

Reply: Exactly, right. Framboids are not always composed of magnetite. For example, the Earth also has iron sulfide framboids. Therefore, “nonmagnetic framboids” does not mean that “framboidal magnetite without strong magnetics.”

– L41: to my knowledge, no paleomagnetic study of framboid-bearing extraterrestrial samples has unequivocally shown that framboids carry a non-zero magnetization. I would not use “normally”.

Reply: “normally” has been deleted.

– L45: recorded a magnetic field, not acquired.

Reply: “acquired” has been changed to “recorded”.

– L54: traces of what?

Reply: “space weathering” has been added.

– How can an examination of the “traces” provide insight on interplanetary processes? Can you be more explicit?

Reply: The study of traces of space weathering of extraterrestrial material provides clues about the formation, evolution, and current state of solar-system bodies. In this perspective, the following examples were added at the end of the text: “such as relative ages of surfaces on airless bodies and accurate interpretation of remote

P

A

G

E

sensing data.”

– Aren’t meteorites subjected to space weather after their excavation event?

Reply: After meteorites are excavated, any traces of space weathering before they come to Earth are lost due to heating as they enter the atmosphere. Since we are talking here about space weathering effects on the surface of asteroids, we do not discuss space weathering after they become meteorites.

– What is the lithification of the regolith breccias?

Reply: The lithification of regolith breccias refers to the process by which unconsolidated materials (regolith) on the surfaces of asteroids or other celestial bodies are transformed into solid rock (breccias) through compaction and cementation.

– L92: remove “as its name suggests”

Reply: Removed.

– L92-94 I don’t understand this sentence

Reply: To clarify the sentence, it has been rewritten as “Magnetite, a ferromagnetic mineral, is important as the major carrier of remanent magnetization.” The explanation can be found in the subsequent sentences.

– L97: this is IF the aqueous alteration occurs in the lifetime of the solar nebula field.

Reply: “nebula” has been deleted.

– L98-100: Note that in a recent publication, we propose a different interpretation of their results in light of measurements we conducted on other Ryugu samples. We do not find any stable magnetization in our 3 samples and argue that the results of Nakamura et al and Sato et al are due to magnetic contamination. This might affect some of your discussions in this manuscript. Ref: Maurel, C., J. Gattacceca, and M. Uehara. 2024. “Hayabusa 2 Returned Samples Reveal a Weak to Null Magnetic Field during Aqueous Alteration of Ryugu’s Parent Body.” Earth and Planetary

P

A

G

E

Science Letters 627 (February): 118559.

Reply: Following sentence has been added “**On the other hand, no stable magnetization was observed in other three different Ryugu samples (Maurel et al. 2024).**”

– L110: remove “surrounding”

Reply: Removed.

– Table S1: what are the units? Are these normalized total? If so, can you indicated the non-normalized total ?

Reply: The unit is atomic %, which has been added in Tagle S1. Elemental compositions obtained by EDS are always relative values and therefore cannot represent un-normalized totals.

– L185: why is the Ni content not so well constrained?

Reply: The iron particles formed by leaching from magnetite do not contain nickel because magnetite does not contain nickel. On the other hand, the original iron particles that were present on the asteroid do contain nickel. The nickel content is due to this difference.

– L190-194 and associated figure: hard to understand this part.

Reply: We hope following explanation helps to understand: The iron particles found inside have between 8% to 12% nickel, while those in the middle area have about 4% nickel. This difference might be because the iron reacted with nickel around it. When we look at the patterns of iron and nickel from the outer layer to the inside, they match up with the idea that this happened because of the heat from impacts. Corresponding sentences have been revised as below: “The presence of many tiny iron particles **without nickel near the surface of the region in Fig. 5C suggests that the tiny nickel-free** iron particles formed as a result of the reduction of magnetite by micrometeoroid bombardment (Fig. 4 and SI, Table S1). **Since the iron particles in the interior contained 8–12 at% Ni, iron particles containing ~4 at% Ni in the intermediate region may be a result of reaction with the iron-nickel particles in the interior.** Line profiles of iron and nickel from the surface to the interior are also

P

A

G

E

consistent with this formation scenario involving impact heating (SI, Fig. S5).”

– L201: reference to back up that 10^4 particles are enough?

Reply: We added the description concerning statistical thermodynamics: “In terms of statistical thermodynamics, the paleomagnetic samples should contain a sufficiently large numbers of ferromagnetic grains to obtain accurate paleomagnetic data (e.g., Berndt et al., 2016). An X-ray photoemission electron microscopy study on the Imilac and Esquel pallasite meteorites demonstrated that the regions containing ~10⁴ grains recorded the systematic changes in a dynamo magnetic field (Bryson et al., 2015), whereas the estimation of the statistical error of paleomagnetism assuming specific conditions showed that the analyzed regions in these meteorites were not large enough to obtain accurate paleomagnetic data (Berndt et al., 2016).”

– L235-237: where are the associated data?

Reply: New figure, Fig. S9, has been added.